# Improving LLMs' Generalized Reasoning Abilities by Graph Problems

**Qifan Zhang**[*]  **Nuo Chen**[*]  **Zehua Li**   **Miao Peng**   **Jing Tang**   **Jia Li**[†]
The Hong Kong University of Science and Technology (Guangzhou)
{qzhang297, zli243, mpeng885}@connect.hkust-gz.edu.cn
chennuo26@gmail.com, {jingtang, jialee}@ust.hk

## Abstract

Large Language Models (LLMs) have made remarkable strides in reasoning tasks, yet their performance often falters on novel and complex problems. Domain-specific continue-pretraining (CPT) methods, such as those tailored for mathematical reasoning, have shown promise but lack transferability to broader reasoning tasks. In this work, we pioneer the use of Graph Problem Reasoning (GPR) to enhance LLMs' general reasoning capabilities. GPR tasks—spanning pathfinding, network analysis, numerical computation, and topological reasoning—require sophisticated logical and relational reasoning, making them ideal for teaching diverse reasoning patterns. To achieve this, we introduce GraphPile, the first large-scale corpus specifically designed for CPT using GPR data. Spanning **10.9** billion tokens across **23** graph tasks, the dataset includes **Chain-of-Thought**, **Program-of-Thought**, **Trace of Execution**, and **Real-world Graph Data**. Using GraphPile, we train GraphMind on three popular base models-Llama 3&3.1 and Gemma 2-achieving up to 4.9% higher accuracy in mathematical reasoning and up to 21.2% improvement in non-mathematical reasoning tasks, like logical and commonsense reasoning. By being the first to harness GPR for enhancing reasoning patterns and introducing the first dataset of its kind, our work bridges the gap between domain-specific pretraining and universal reasoning capabilities, advancing the adaptability and robustness of LLMs.

## 1 Introduction

Recent advancements in Large Language Models (LLMs) have demonstrated impressive few-shot learning capabilities across a wide range of tasks (Kirillov et al., 2023; Sun et al., 2024; You et al., 2022a; Cobbe et al., 2021a; Zhou et al., 2022; Wei et al., 2022; Wang et al., 2022; Chen et al., 2023a; You et al., 2022b; Chen et al., 2024b). However, when faced with novel and complex problems, their performance often falls short. To address this limitation, many works have focused on continue-pretraining (CPT), particularly in the domain of mathematical reasoning (Shao et al., 2024; Azerbayev et al., 2023; Ying et al., 2024; Yang et al., 2024; Wang et al.; Gunasekar et al., 2023; Lu et al., 2024). A variety of models have emerged from this paradigm, such as MathCoder (Wang et al., 2023b), Qwen-Math (Yang et al., 2024), and the DeepSeek Math series (Shao et al., 2024). These models leverage vast datasets focused on mathematical problems or code-related tasks, often gathered from web sources (Shao et al., 2024; Yang et al., 2024) or synthetically generated (Gunasekar et al., 2023; Yang et al., 2024), to improve the performance of LLMs in solving mathematical challenges.

While these efforts have yielded significant improvements in mathematical reasoning, they are primarily domain-restricted (Wang et al., 2023c; Wu et al., 2023). The question remains whether such focused pretraining can translate to broader reasoning capabilities. In particular, the improvements observed in mathematical tasks have not been demonstrated to extend to other complex reasoning areas, such as algorithmic understanding and logical

---

[*] Equal Contribution
[†] Corresponding author

| Benchmark | Logic | Topology | Computation | Enumeration | Division |
|-----------|-------|----------|-------------|-------------|----------|
| Math Reasoning | ✓ | ✗ | ✓ | ✓ | ✓ |
| - GSM8K | ✓ | ✗ | ✓ | ✗ | ✓ |
| - MATH | ✓ | ✗ | ✓ | ✓ | ✓ |
| - AIME-2024 | ✓ | ✗ | ✓ | ✓ | ✓ |
| - MMLU-STEM | ✓ | ✗ | ✓ | ✓ | ✓ |
| **Algorithm Reasoning** | ✓ | ✓ | ✓ | ✓ | ✓ |

(a) Reasoning paradigm comparison

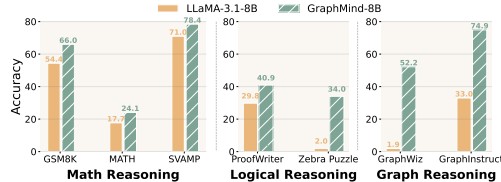

(b) GraphMind-LLaMA-3.1 results

Figure 1: Reasoning paradigm comparison between math and graph (left) and overview of GraphMind-LLaMA-3.1 results compared with baseline models on math, logical and graph reasoning benchmarks (right).

reasoning. This leads us to explore a fundamental question: Rather than being domain-locked, is there a class of problems or data that, if incorporated into LLM training, could foster the model's general reasoning ability across diverse tasks?

Graph Problem Reasoning (GPR) (Chen et al., 2024a; Tang et al., 2024; Zhang et al., 2024b; Peng et al., 2025) presents a compelling answer to this question. GPR involves tackling challenges rooted in graph theory—a foundational area of mathematics concerned with the study of graphs, which consist of nodes (vertices) and edges (connections between nodes). The domain of graph problems spans a wide variety of complex tasks, including pathfinding, network analysis, and edge counting, all of which require sophisticated multi-step reasoning and an ability to navigate intricate, relational structures. What makes GPR particularly intriguing is its resemblance to mathematical reasoning, as shown in Figure 1 (a). Both domains share several key characteristics, such as the need for logical computation, systematic enumeration, and division awareness. For instance, in mathematical problem solving, a series of well-defined steps is often required to reach a solution. Similarly, graph-based tasks necessitate a stepwise approach to traverse, analyze, or optimize the graph structure. Notebaly, the reasoning patterns involved in the design of currently popular mathematical datasets are all a subset of GPR, like GSM8K (Cobbe et al., 2021b) and MATH (Hendrycks et al., 2021b).

In general, GPR tasks demand different reasoning patterns, including topological reasoning, logical reasoning, enumeration, precise computation and division (More details in Sec.2.1), making them fundamentally challenging for LLMs (Fatemi et al., 2023; Wang et al., 2023a). In many ways, solving these problems mirrors the process of mathematical reasoning, which also relies on logical steps to arrive at a solution. Moreover, GPR introduces unique challenges not present in traditional mathematical reasoning. For instance, while mathematical problems typically involve direct mathematical formulas and manipulations, graph problems often involve relational reasoning—understanding how entities (nodes) are connected and how their relationships impact the overall solution. This introduces additional complexity, as the reasoning must account for the spatial or topological relationships between nodes and edges, something that is less pronounced in classical mathematical problem-solving. Meanwhile, the difficulty of graph problems often scales exponentially with the size and complexity of the graph (Tang et al., 2024), posing an additional layer of challenge that further tests the limits of an LLM's reasoning capacity.

Given the intrinsic similarities between mathematical reasoning and graph problem-solving, and the wide-ranging nature of graph-based tasks, we hypothesize that graph problem reasoning can serve as an effective source of data for continue-pretraining LLMs. By integrating graph problems into the pretraining process, we aim to unlock a powerful tool for enhancing general reasoning abilities. Specifically, we explore the potential of GPR to improve not only mathematical reasoning but also other forms of complex reasoning, such as algorithmic problem-solving and logical reasoning. We seek to bridge the gap between domain-specific pretraining and the development of more universally capable reasoning models, ultimately making LLMs more robust, adaptable, and effective in a broader range of problem-solving scenarios.

In this work, our goal is to include graph problem reasoning data that helps LLMs evolve into more generalized and better reasoners. To this end, we present the first dataset for

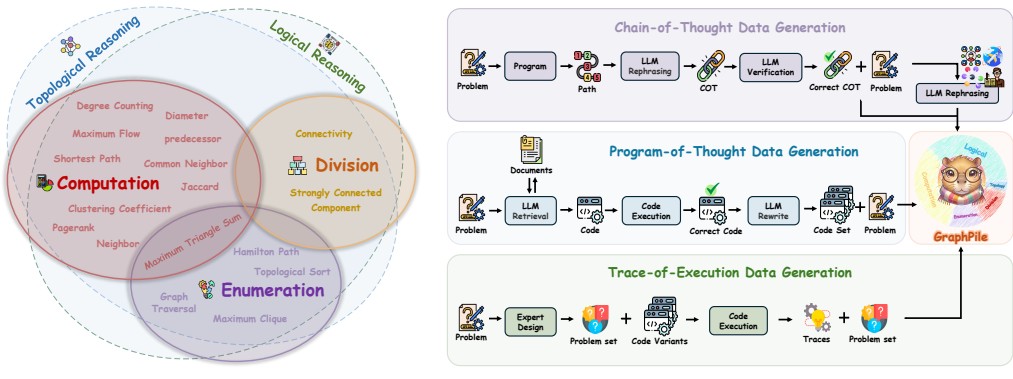

(a) Different reasoning paradigms  (b) Dataset curation pipeline

Figure 2: Overall illustration of (a) Different reasoning paradigms for total 23 graph problem tasks in `GraphPile`, and (b) `GraphPile` curation pipeline.

continue-pretraining LLMs using Graph Problem Reasoning data, named `GraphPile`, consisting of approximately **10.9** billion tokens and covering **23** distinct graph problem tasks. Our dataset includes four main types of data: 1) **Chain-of-Thought Data** (Wei et al., 2022; Yao et al., 2023): For each graph problem, we employ a program-guided approach to generate accurate chain-of-thought (CoT) solutions, enabling the model to learn the reasoning process step by step; 2) **Program-of-Thought Data** (Chen et al., 2022): Given a graph problem, we leverage LLMs to identify relevant code repositories from web text and generate corresponding solutions, thereby connecting problem-solving to code generation; and 3) **Trace of Execution Data**: A novel data type introduced in this work, where we record the execution trace of graph problem algorithms, allowing the model to learn from the execution process itself, potentially enhancing its ability to understand algorithmic steps and reasoning; 4) **Real-world Graph Data**: In addition to these synthetic graph problems, we also collect real-world graph problems from sources like DBpedia (Bizer et al., 2009) and DBLP (Ley, 2002), further enriching our dataset with practical, domain-specific problems that reflect real-world graph analysis challenges.

We train three popular base models on `GraphPile`, including LLaMA-3&3.1 8B (Grattafiori et al., 2024) models and Gemma-2-2B models (Team et al.), resulting in `GraphMind` series. We validate the effectiveness of our models across 5 distinct reasoning tasks and 20 datasets, achieving significant improvements in all cases, as shown in Figure 1 (b). We primarily consider two settings for evaluation: 1) Few-shot: We directly test `GraphMind` across different datasets using few-shot prompts; 2) Post-training: We fine-tune `GraphMind` on downstream datasets when training sets are available, further refining its performance in specific tasks. Experimental results indicate that `GraphMind` exhibits enhanced reasoning capabilities across various domains compared to the base model. In mathematical reasoning, its average accuracy over 11 datasets surpasses the base model by up to 4.9%, while in other reasoning tasks, the improvement reaches as high as 21.2%. Moreover, `GraphMind` demonstrates potential for post-training enhancement. Notably, the Gemma version of `GraphMind` fine-tuned on GSM8K achieves 23.6% higher accuracy.

## 2  `GraphPile`

In this section, we present the curation details of `GraphPile`, covering reasoning paradigms definitions (Section 2.1) and the dataset curation pipeline (Section 2.2). An overview is in Figure 2.

### 2.1  Reasoning Paradigms

Graph problem reasoning cover various reasoning paradigms, including logical reasoning, topological reasoning, numerical computation, enumeration, and division. To ensure

comprehensive coverage, we include tasks from each category. For clarity, we select two representative tasks per category, focusing on their primary reasoning type, though some tasks may overlap categories.

**Logical Reasoning Tasks.** Logical reasoning involves analyzing, deducing conclusions, or solving problems based on *specific logical rules*. **Since all graph problem tasks are fundamentally based on reasoning derived from logical rules, they are inherently tasks of logical reasoning.** Here are two representative tasks.

- **Cycle Detection.** Given a graph $\mathcal{G} = \{\mathcal{V}, \mathcal{E}\}$, the task is to check if it contains a cycle. A cycle is a closed path with at least three vertices, where each pair of consecutive vertices is connected by an edge.

- **Bipartite Checking.** This task is to determine if a graph $\mathcal{G} = \{\mathcal{V}, \mathcal{E}\}$ is bipartite, meaning its vertex set can be partitioned into two subsets **U** and **V** where no two vertices in the same subset are adjacent.

Solving these problems requires LLMs to apply specific logical rules. For example, a cycle exists in a graph if a path from a vertex revisits the same vertex. A graph is bipartite if its vertices can be colored with two colors so that no adjacent vertices share the same color.

**Topological Reasoning Tasks.** Topological reasoning tasks involve exploring the **relationships between nodes and edges** in a graph and making inferences based on these relationships. *Since all graph problem tasks are built upon these relationships, they inherently belong to the domain of topological reasoning.* Representative tasks include topological sorting and common neighbors.

- **Topological Sorting.** For a directed acyclic graph (DAG) $\mathcal{G} = (\mathcal{V}, \mathcal{E})$, a topological sorting is a linear ordering of its vertices such that for every directed edge $(u, v) \in \mathcal{E}$, the vertex $u$ appears before $v$ in the ordering.

- **Common Neighbors.** Given a graph $\mathcal{G} = (\mathcal{V}, \mathcal{E})$, the task is to identify the set of common neighbors between two nodes $u, v \in \mathcal{V}$. The common neighbors are defined as $\mathcal{N}(u) \cap \mathcal{N}(v)$, where $\mathcal{N}(x)$ represents the set of neighbors of node $x$.

Solving these problems typically requires LLMs to understand the topological structure of the graph. For example, topological sort reveals hierarchical relationships in directed acyclic graphs, while common neighbors highlight local connections between two nodes. This reasoning paradigm is rarely encountered in traditional mathematical problems.

**Numerical Computation Tasks.** Numerical computation involves using algorithms to solve problems through **a large number of operations** like addition, subtraction, multiplication, and division. Representative tasks are the shortest path and maximum flow.

- **Shortest Path.** Given a weighted graph $\mathcal{G} = \{\mathcal{V}, \mathcal{E}, w\}$ with $w : \mathcal{E} \to \mathbb{R}^+$, the task is to find a path between two nodes that minimizes the total edge weight.

- **Maximum Flow.** Given a weighted graph $\mathcal{G} = \{\mathcal{V}, \mathcal{E}, c\}$ with capacities $c : \mathcal{E} \to \mathbb{R}^+$, a source $v_s$, and a sink $v_t$, the task is to maximize the flow from $v_s$ to $v_t$.

Solving these problems requires LLMs to perform numerical computations. For example, the shortest path problem involves iterative updates of distances, and the maximum flow problem requires tracking residual capacities. Math problems like root finding and numerical integration are also examples of numerical computation.

**Enumeration Tasks.** Enumeration tasks involve systematically listing all possible solutions or elements in a set, often to address problems in **combinatorics, optimization, or search**. Representative examples include the Hamilton path and the maximum matching.

- **Hamilton Path.** Determine if a Hamilton path exists in graph $\mathcal{G} = (\mathcal{V}, \mathcal{E})$, where a Hamilton path visits each node exactly once.

- **Maximum Clique Problem.** The task is to find a clique of maximum size in a graph $\mathcal{G} = \{\mathcal{V}, \mathcal{E}\}$. A clique is defined as a subgraph where all nodes are fully connected.

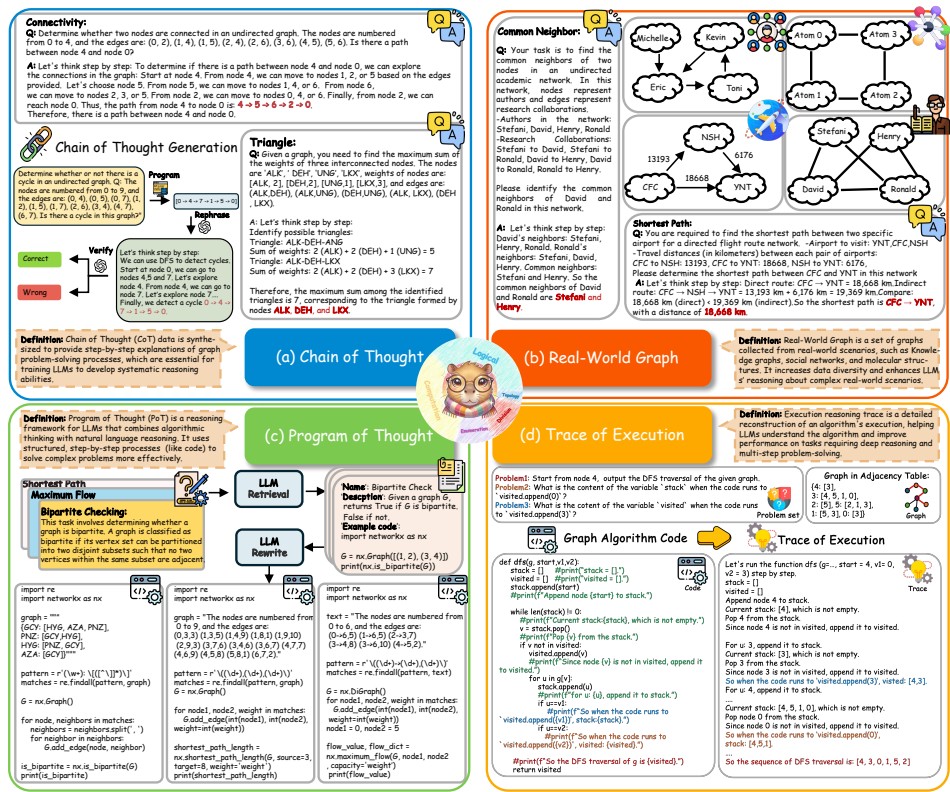

Figure 3: Data components of GraphPile.

Solving these problems often requires LLMs to enumerate the solution space. For the Hamilton Path problem, this involves listing paths that visit each node exactly once, while the Maximum Clique problem requires enumerating cliques. Similarly, math problems like the integer partition problem and subset sum problem are examples of enumeration.

**Division Tasks.** Division tasks focus on **breaking down a large problem into smaller, independent subproblems**, which can be solved individually and then combined to form the overall solution. These tasks are commonly encountered in areas such as connectivity and strongly connected components in graph theory.

- **Connectivity.** The task is to determine if two nodes $u$ and $v$ in a graph $\mathcal{G} = (\mathcal{V}, \mathcal{E})$ are connected, meaning there exists a path from $u$ to $v$.
- **Strongly Connected Component.** The task is to find the Strongly Connected Components (SCCs) of a directed graph $\mathcal{G} = (\mathcal{V}, \mathcal{E})$, where an SCC is a maximal subgraph in which every pair of nodes $u$ and $v$ has paths both from $u$ to $v$ and from $v$ to $u$.

Both tasks require LLMs to employ the division approach to systematically decompose a graph into several connected components. Mathematical problems such as modular arithmetic and polynomial division also fall under this category.

In addition to the aforementioned reasoning paradigm, *Graph Problem Reasoning also encompasses spatial reasoning, such as planarity testing*. To enhance the diversity of our dataset, we have selected a total of **23** graph problem tasks, as demonstrated in Figure 2 (a). Detailed task definitions are provided in Appendix B.1.

## 2.2 GraphPile Collection

In this section, we introduce the curation process of GraphPile, a diverse collection of graph problem data for continued pretraining LLMs. Our data synthesis consists of four

components: **Chain-of-Thought data**, **Real-World Graph data**, **Program-of-Thought data**, and **Trace of Execution Data**. Since our synthesis process involves various problems related to graph algorithms, *we hire a human expert* [1] *in this field to design algorithm implementations, verify correctness, and optimize efficiency*. Figure 2 (b) provides an overview of these pipelines.

**Graph Generation.**  `GraphPile` comprises both synthetically generated Erdős-Rényi (ER) (Erdős et al., 1960) graphs and real-world graphs obtained through web crawling. For each type of graph, we include both directed and undirected variants, which are randomly generated. Furthermore, the dataset encompasses a variety of graph representations, including adjacency matrices, adjacency lists, and edge lists. We include graphs of varying sizes, ranging from 6 to 40 nodes, for the following reasons: (1) To address varying levels of complexity in graph reasoning; and (2) To limit graph sizes, as larger graphs result in extremely large sample length (more than 8k) and make it harder for LLMs to learn accurate and complete reasoning path during continue-pretraining.

**Chain-of-Thought Data Generation.**  CoT data is synthesized to offer step-by-step explanations of graph problem-solving processes, which are essential for training LLMs to develop systematic reasoning abilities. However, graph problem reasoning differs significantly from mathematical reasoning because its difficulty increases exponentially as the graph size grows, making it a major challenge even for state-of-the-art LLMs such as GPT-4o. As a result, LLMs struggle to directly generate accurate and complete CoT solutions, especially for large and complex graphs (Chen et al., 2024a; Luo et al., 2024). To address this, we propose a **Program-Guided approach**: for each graph problem, an expert designs a corresponding programmatic solution. By executing this program, we can record intermediate computational results. For example, in the cycle detection problem illustrated in Figure 3 (a), we log the path leading to the final decision, such as 0-4-7-1-5-0. These paths are then rephrased into natural language solutions using GPT-4o. However, due to the inherent variability in LLM outputs, we employ GPT-4o again to verify the correctness of the rephrased paths. The correctly rephrased paths, together with the original graph problems, form our CoT dataset. Besides, since many graph problems admit multiple valid solutions (e.g., in cycle detection, multiple cycles may exist in a given graph), we sample up to three distinct solutions to enhance the diversity of CoT reasoning data. Detailed prompt of LLM rephrasing and LLM verification are given in Appendix C.1.

**Real-World Graph Data Generation.**  To enhance the diversity of Chain-of-Thought data and improve the reasoning capabilities of LLMs in complex realistic scenarios, we extend our CoT dataset to include real-world graph problems. Specifically, we begin by collecting real-world graphs from sources such as DBLP (Ley, 2002), OpenFlight (OpenFlights), PubChemQC (Nakata & Shimazaki, 2017), Social Networks (Rossi & Ahmed, 2015), and DBPedia (Bizer et al., 2009). We then utilize GPT-4o to rephrase the problems and answers from the existing CoT dataset by replacing numerical node identifiers with real-world textual node identifiers. After rephrasing, we use GPT-4o to filter out incorrect problem and answer pairs (e.g., cases where nodes or edges in the original graph are lost or added during the rephrasing process). Through this process, we construct the **Real-World Graph** dataset, which features real-world problems paired with step-by-step solutions. This dataset effectively bridges abstract graph reasoning with practical, real-world applications. Figure 3 (b) provides examples of this dataset. Related prompts are given in Appendix C.2.

**Program-of-Thought Data Generation.**  While CoT provides step-by-step explanations, Program-of-Thought (PoT) offers precise, executable solutions that eliminate ambiguity (Li et al., 2024b;a; Zhang et al., 2024a; Wang et al., 2024b). Training LLMs on well-structured code enhances their ability to interpret, generate, and apply algorithmic logic effectively in graph problem-solving. Considering that many graph-related algorithms have already been implemented in widely used libraries such as NetworkX, we leverage these resources for PoT solution generation. After constructing a graph problem, we prompt the LLM to retrieve relevant implementations directly from code documentation repositories. If

---

[1]This expert holds a PhD specializing in graph algorithms.

| Dataset | Graph Category | Problem-Solving Paradigms | Tasks | Samples | CPT-Compatible |
|---|---|---|---|---|---|
| GraphWiz (Chen et al., 2024a) | Synthetic | CoT | 9 | 17,158 | ✗ |
| GraphInstruct (Luo et al., 2024) | Synthetic | CoT | 21 | 16,800 | ✗ |
| InstructGraph (Wang et al., 2024a) | Synthetic | Simple Answer | 6 | 13,699 | ✗ |
| GraphArena (Tang et al., 2024) | Real-World | Simple Answer | 10 | 10,000 | ✗ |
| **GraphPile (Ours)** | **Synthetic + Real-World** | **CoT, PoT, ToE** | **24** | **2,684,675** | ✓ |

Table 2: Comparison between GraphPile and existing graph reasoning datasets

the extracted code is executable and produces correct results, we include it in our dataset. Furthermore, to ensure PoT diversity, we instruct the LLM to rewrite the extracted code, generating a modified version that retains the original logic while introducing variations in implementation. These alternative versions help LLMs generalize across different coding styles and improve their robustness in algorithmic reasoning. Figure 3 (c) demonstrates examples of this dataset. Related prompts are given in Appendix C.3.

**Trace-of-Execution Data Generation.** Execution reasoning trace is a textual sequence that explains the step-by-step execution of a code, abstracting the reasoning process while maintaining logical rigor. It captures key skills like logical flow, state exploration, recursion, and decision-making. However, this data paradigm has not been studied before. To leverage its benefits, we propose the **Program-Guided** approach. For a given graph problem, experts design

| Components | Size | Tokens |
|---|---|---|
| Chain-of-Thought | 848,965 | 2,809,225,185 |
| Real-world Graph | 743,465 | 3,203,590,685 |
| Program-of-Thought | 759,851 | 2,190,746,959 |
| Trace-of-Execution | 332,394 | 2,727,119,224 |
| **Total** | **2,684,675** | **10,930,682,053** |

Table 1: Statistics of GraphPile.

three versions of high-quality code with tracing phrases at key points. Each code is along with two additional problems focusing on intermediate variable states, forming a diverse problem set. Executing the codes generates traces, which, combined with the problems, form the Trace of Execution data, enabling LLMs to learn from multiple reasoning perspectives. For example, Figure 3 (d) shows a graph traversal problem, its solution code, and tracing phrases at key points. Two extra problems focus on the intermediate states of "stack" and "visited". By executing the code, the solution trace is generated.

Finally, by integrating these datasets, we construct our training dataset, `GraphPile`, which comprises 4 distinct components and encompasses over **2.68** million samples and **10.9** billion tokens. `GraphPile` integrates different reasoning paradigms, empowering LLMs to tackle graph problems with greater robustness, systematic precision, and efficiency. Detailed statistics of `GraphPile`, the comparison between `GraphPile` and existing graph reasoning datasets, and more examples of each dataset are given in Table 1, Table 2 and Appendix D.1.

## 3 Experiments

After constructing `GraphPile`, We begin by performing continue-pretraining on three LLMs, ranging in size from 2B to 8B, thereby obtaining three versions of our reasoning model, `GraphMind`. In this section, we validate our model in two scenarios: 1) **Few-shot**: We directly test `GraphMind` across different datasets with few-shot prompts; 2) **Post-training**: We fine-tune `GraphMind` in downstream datasets if training sets are available. At last, we perform an ablation study to evaluate the contribution of different components within `GraphPile`.

### 3.1 Experimental Settings

**Evalution Dataset.** To thoroughly evaluate the reasoning capabilities of `GraphMind` and other baselines, we select **22** benchmarks spanning six reasoning domains: **mathematics reasoning**, **logical reasoning**, **commonsense reasoning**, **code reasoning**, **multi-Hop QA reasoning**, and **graph problem reasoning**. For mathematics reasoning benchmarks, we choose 11 benchmarks, including GSM8K (Cobbe et al., 2021b), MATH (Hendrycks et al., 2021b), GSM8K-Hard (Gao et al., 2022), SVAMP (Patel et al., 2021), ASDIV (Miao et al., 2020), MAWPS (Koncel-Kedziorski et al., 2016), MINERVA_MATH (Hendrycks et al., 2021b),

MMLU_STEM (Hendrycks et al., 2021a)), TABMWP (Lu et al., 2022), MATHQA (Amini et al., 2019) and SAT_Math (Zhong et al., 2023). For logical reasoning benchmarks, we choose Zebra Puzzle (Lin et al., 2025), Ruletaker (Clark et al., 2020), and ProofWriter (Tafjord et al., 2020). For commonsense reasoning, we choose Strategy QA (Geva et al., 2021) and Hellaswag (Zellers et al., 2019). For code reasoning, we choose Livecodebench (Jain et al., 2024) and CLRS (Markeeva et al., 2024). For multi-hop QA reasoning, we choose HotpotQA (Yang et al., 2018) and PopQA (Mallen et al., 2022). For GPR benchmark, we choose GraphWiz (Chen et al., 2024a) and GraphInstruct (Luo et al., 2024). We collect these benchmarks and utilize them to evaluate the models based on three GitHub projects: OpenCompass (Contributors, 2023), Qwen2.5-Math (Yang et al., 2024), and ZeroEval (Lin, 2024). **See Appendix E.1 for details of base models, training and evaluation settings, and E.2 for details of evaluation datasets.**

| Models | Mathematics | | | | | | | | | | | |
|---|---|---|---|---|---|---|---|---|---|---|---|---|
| | GSM8K | MinMath | MATH | GSM-Hard | SVAMP | ASDIV | MAWPS | STEM | TABMWP | MATHQA | SAT | Avg. |
| Gemma-2-2b | 26.9 | 14.4 | 13.2 | 16.8 | 48.7 | 62.3 | 77.6 | **43.6** | 39.6 | 32.8 | **53.1** | 39.0 |
| + GraphPile | **36.8** | **15.0** | **18.7** | **18.5** | **58.6** | **66.0** | **83.9** | 41.9 | **42.0** | **38.0** | 40.6 | **41.8** |
| Llama-3-8b | 54.2 | 17.0 | 16.5 | 26.1 | 68.8 | 73.1 | 90.9 | 49.7 | **57.9** | 27.7 | **56.2** | 48.9 |
| + GraphPile | **65.8** | **21.6** | **24.0** | **29.3** | **78.9** | **79.5** | **91.6** | **56.1** | 50.6 | **41.0** | 53.1 | **53.8** |
| Llama-3.1-8b | 54.4 | 20.4 | 17.7 | 27.1 | 71.0 | 74.3 | 92.0 | **57.0** | 63.6 | 44.9 | 59.4 | 52.9 |
| + GraphPile | **66.0** | **23.6** | **24.1** | **30.9** | **78.4** | **79.6** | **92.0** | 56.4 | 59.2 | **53.1** | 59.4 | **56.6** |

| Models | Logic | | | Commonsense | | Code | | Multi-Hop QA | | Graph Problem | | |
|---|---|---|---|---|---|---|---|---|---|---|---|---|
| | Zebra Puzzles | Ruletaker | ProofWriter | StrategyQA | Hellaswag | LCB | CLRS | HotpotQA | PopQA | GraphWiz | GraphInstruct | Avg. |
| Gemma-2-2b | **2.0** | 3.4 | **8.8** | 59.3 | 27.2 | 0.2 | 16.9 | 21.2 | 27.4 | 34.6 | 16.8 | 19.8 |
| + GraphPile | 0.5 | **20.5** | 6.0 | **59.4** | **30.6** | **1.3** | **37.9** | **41.0** | **28.2** | **54.8** | **62.8** | **31.2** |
| Llama-3-8b | 10.0 | 22.8 | 25.0 | 66.2 | 48.3 | 2.9 | 3.3 | 25.8 | 24.6 | 4.5 | 35.2 | 24.4 |
| + GraphPile | **24.0** | **43.1** | **41.6** | **69.7** | **52.8** | **6.9** | **49.9** | **26.0** | **32.2** | **49.9** | **70.8** | **42.4** |
| Llama-3.1-8b | 18.0 | 35.3 | 29.8 | 58.9 | 48.7 | 1.6 | 3.3 | 43.6 | 40.0 | 1.9 | 33.0 | 28.6 |
| + GraphPile | **34.0** | **61.1** | **40.9** | **69.6** | **53.6** | **12.5** | **5.6** | **46.4** | **47.0** | **52.2** | **74.9** | **45.1** |

Table 3: Main results on mathematical benchmarks (top) and other benchmarks (bottom), where LCB represents Livecodebench.

## 3.2 Main Results

We evaluate the performance of GraphMind and baseline models on 20 benchmarks across four reasoning domains. From Table 3, we can draw an overall conclusion: leveraging GraphPile enhances both foundational reasoning capabilities and generalization reasoning abilities. Regarding foundational reasoning capabilities, when trained on graph problem reasoning data, the three versions of GraphMind achieve a best average improvement of 53.1% compared to the baseline, demonstrating the effectiveness of GraphPile in improving the graph problem reasoning abilities of LLMs. In terms of generalization capabilities, the improvements are as follows: for math reasoning, the best average improvement is 4.9%; for logical reasoning, it is 33.4%; for commonsense, it is 7.8%; for code reasoning, it is 46.3%; and for multi-hop QA, it is 10.3%. As illustrated in Figure 2 (a), graph problem reasoning includes various reasoning patterns. This is why our constructed graph problem reasoning dataset can enhance model performance not only on graph-related tasks but also on mathematical and other reasoning tasks, thereby improving the model's overall generalization ability. The detailed results for each task in the GPR dataset are provided in Table 7 in Appendix E.3. Furthermore, we conduct statistical significance tests in Appendix E.4.

## 3.3 Post-Training

To show the performance improvements of GraphMind through post-training, we select three versions of GraphMind , along with their base models for fine-tuning on GraphWiz, GSM8K, and MATH.

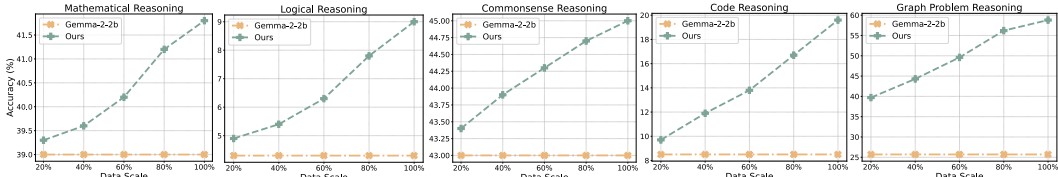

Figure 4: Model performances with scaling `GraphPile` size. Here, we report average results of different reasoning tasks.

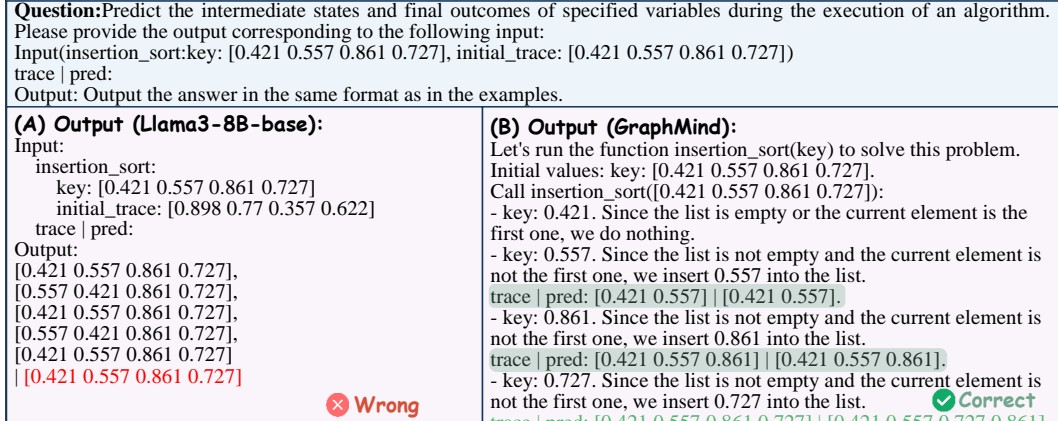

Figure 5: A comparative case between Llama3 and ours in CLRS dataset.

Table 4 shows that `GraphMind` outperforms base models in both graph reasoning and math tasks across all configurations. Notable performance gains are observed in smaller models like Gemma-2, which shows an increase of around 7 points in graph reasoning. Similarly, `GraphMind` demonstrates substantial improvements in mathematical tasks, with the highest performance improvement of 9.7% for Llama-3-8b on GSM8K and 2.4% for Llama-3.1-8b on MATH. The results highlight that `GraphPile` enhances LLMs' post-training potential.

| Models | GraphWiz | GSM8K | MATH |
|---|---|---|---|
| Gemma-2-2b | 34.9 | 37.4 | 12.4 |
| + GraphPile | **42.0** | **61.0** | **16.2** |
| Llama-3-8b | 32.2 | 67.4 | 18.6 |
| + GraphPile | **36.3** | **77.1** | **20.4** |
| Llama-3.1-8b | 44.1 | 71.7 | 21.6 |
| + GraphPile | **53.8** | **73.0** | **24.0** |

Table 4: Comparison between post-training versions of base models and CPT models.

### 3.4 Ablation Studies

We further conduct ablation studies to explore the model performances: (1) when scaling the training dataset size, (2) when removing different components, and (3) when conducting continue-pretraining on stronger base LLMs.

**Scaling the Training Dataset Size.** We sample 20% to 100% of the data from `GraphPile` to continue-pretrain `GraphMind`, resulting in five variants. These variants, along with Gemma-2-2b, are tested on evaluation datasets grouped by mathematical reasoning, logical reasoning, commonsense reasoning, code reasoning, and graph problem reasoning. The results for each group are averaged to represent the model's performance on each type of reasoning dataset. The experimental results in Figure 4 demonstrate that (1) when the amount of data is 20%, `GraphMind` performs similarly to its base model, Gemma-2-2b, in other reasoning domains beyond graph problem reasoning, as the small amount of data is insufficient for the model to learn generalized reasoning capabilities; (2) as the data scale increases, `GraphMind` shows better performance on all reasoning paradigm and gradually surpasses the base model. This illustrates the effectiveness of `GraphPile` and the scalability of `GraphMind`.

**Removing Different Components.** We remove different components of GraphPile to create four reduced datasets: w/o CoT, w/o PoT, w/o RW, and w/o ToE. Using these four datasets, as well as the full dataset, we train GraphMind. Table 5 presents the experimental results. Overall, we observe that the absence of data from a specific domain generally leads to performance degradation.

| Models | Math | GPR | Logical | Code | Com.S |
|--------|------|-----|---------|------|-------|
| GraphMind | 41.8 | **58.8** | 9.0 | **19.9** | 45.0 |
| w/o CoT | 41.0 | 49.8 | 5.6 | 17.9 | 44.7 |
| w/o PoT | 40.8 | 56.3 | 6.7 | 16.2 | **45.6** |
| w/o RW | **41.9** | 57.3 | **9.1** | 19.5 | 43.1 |
| w/o ToE | 40.2 | 53.9 | 4.8 | 9.7 | 44.9 |

Table 5: Performances when removing different data types. Com.S, RW refer to commonsense reasoning and Real-World Graph Data.

Interestingly, we also find that the w/o RW dataset improves performance on mathematical and logical reasoning tasks. This is likely because these two domains lack real-world scenarios, making real-world data less beneficial for training the model in these areas.

| Model | Mathematical Reasoning | | | | Logical Reasoning | | | Graph Problem Reasoning | | |
|-------|------|-----------|-----|------|-------------|----------|------|----------|--------------|------|
| | GSM8K | MMLU-STEM | SAT | Avg. | Zebra Puzzle | KorBench | Avg. | GraphWiz | GraphInstruct | Avg. |
| Qwen-2.5-Coder-1.5B | 59.8 | 32.9 | 59.4 | 50.7 | 1.9 | 16.8 | 9.4 | 30.3 | 25.1 | 27.7 |
| + GraphPile | **63.4** | **43.3** | **71.9** | **59.5** | **5.7** | **18.3** | **12.0** | **48.6** | **46.1** | **47.4** |
| Qwen-2.5-Coder-7B | 77.9 | 67.2 | 81.2 | 75.4 | 3.9 | 32.1 | 18.0 | 38.5 | 34.4 | 36.5 |
| + GraphPile | **81.0** | **68.3** | **87.5** | **79.0** | **4.8** | **33.3** | **19.1** | **54.3** | **50.3** | **52.3** |

Table 6: Performance of Qwen-2.5-Coder Models with and without GraphPile on Mathmatical Reasoning, Logical Reasoning, and Graph Problem Reasoning Benchmarks.

**Continue-pretraining on Stronger LLMs.** We further conduct continue-pretraining on two stronger base models, Qwen-2.5-Coder-1.5B and Qwen-2.5-Coder-7B, to assess the effectiveness of GraphPile on more capable architectures. We evaluate these base models across three categories of reasoning benchmarks: (1) mathematical reasoning (GSM8K, MMLU-Stem, and SAT), (2) logical reasoning (Zebra Puzzle and KorBench), and (3) graph problem reasoning (Graphwiz and GraphInstruct). The evaluation results are summarized in Table 6. Our analysis demonstrates that continue-pretraining on GraphPile leads to performance improvements across all evaluation domains—including both in-domain (graph reasoning) and out-of-domain (mathematical and logical reasoning) tasks—for all examined models. Specifically, Qwen-2.5-Coder-1.5B achieves improvements of 8.8, 2.6, and 19.7 points over its base model on mathematical, logical, and graph reasoning tasks, respectively. Similarly, Qwen-2.5-Coder-7B exhibits gains of 3.6, 1.1, and 15.8 points across the same task categories. These consistent improvements across different model scales provide compelling evidence that GraphPile enhances performance not only for weaker base models but also for more powerful ones.

**Case Study.** Additionally, we present an example from the CLRS dataset involving an insertion sort problem, where GraphMind produces correct answers while the base model provides incorrect responses, as shown in Figure 5. We observe that, compared to the base model, GraphMind's responses include more detailed intermediate steps—such as highlighting the current sorting element and explaining the rationale behind each insertion—while effectively omitting irrelevant information, such as extraneous input details. Further examples can be found in Appendix E.5.

## 4 Conclusion

We present GraphPile, a 10.9B-token dataset with 23 graph tasks to enhance LLM reasoning. By pretraining three LLMs on it, we develop the GraphMind series, which improves base models by up to 4.9% (math) and 21.2% (other tasks). The Gemma version fine-tuned on GSM8K achieves 23.6% higher accuracy, demonstrating strong post-training potential. Our results show Graph Problem Reasoning effectively boosts LLMs' general reasoning.

## Acknowledgements

This work is supported by NSFC Grant No.62206067. In addition, we express our sincere gratitude to the Baidu Scholarship 2024 for its substantial support in facilitating this research endeavor.

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

## A  Related Works

**LLMs for graph problem reasoning.**  Leveraging large language models (LLMs) for graph problem reasoning has emerged as a prominent area of research. To advance this field, researchers have introduced a variety of benchmarks designed to evaluate the graph problem reasoning capabilities of LLMs (Wu et al., 2024b; Tang et al., 2024; Li et al., 2024a; Das et al., 2023; Fatemi et al., 2023; Guo et al., 2023). Building on these benchmarks, various approaches have been proposed to enhance LLMs' graph problem reasoning capabilities. These methods can be broadly categorized into the following paradigms: (1) Chain-of-Thought (CoT): This paradigm leverages step-by-step reasoning processes to improve LLM performance, as demonstrated in several studies (Chen et al., 2024a; Luo et al., 2024). (2) Program-of-Thought (PoT): This paradigm involves teaching LLMs to generate code for solving graph-related problems (Zhang et al., 2024a; Li et al., 2024b; Wang et al., 2024b; Gong et al., 2025). In this work, we introduce a novel paradigm, **Trace of Execution (ToE)**, which leverages traces—detailed representations of the algorithmic execution process. Our dataset integrates three paradigms—CoT, PoT, and ToE—to enhance the reasoning capabilities of LLMs in graph problems and other domains.

**Continue-Pretraining.** Continue-pertaining methods have been widely used in improving LLMs' reasoning capabilities in specific domains such as mathmatics (Shao et al., 2024; Azerbayev et al., 2023; Ying et al., 2024; Yang et al., 2024; Wang et al.; Gunasekar et al., 2023; Lu et al., 2024), medicine (Chen et al., 2023b; Wu et al., 2024a), and algorithms (Markeeva et al., 2024). By continue-pretraining on a well-curated corpus, the model enhances its ability to handle domain-specific terminology, concepts, and patterns, making it more effective for specialized tasks. For instance, DeepSeekMath (Shao et al., 2024) and Qwen-2.5-Math (Yang et al., 2024) leverage fastText (Joulin et al., 2016) and other meta-information to retrieve texts from the Common Crawl dataset. MathPile (Wang et al.) and Phi (Gunasekar et al., 2023) utilize real or synthesized textbooks to improve LLMs' mathematical reasoning capabilities. MEDITRON (Chen et al., 2023b) employs a meticulously curated medical corpus, including selected PubMed articles, abstracts, and internationally recognized medical guidelines. However, no prior work has focused on utilizing graph problem corpus for pre-training. To fill this gap and further enhance the reasoning capabilities of LLMs, we, for the first time, propose a graph problem-based training corpus aimed at improving LLMs' reasoning abilities across graph-related tasks, mathematics, and other domains.

## B  Additional Definition

### B.1  Task Definitions

In this section, we present the remaining graph problems that were not introduced in Section 2.1.

- **PageRank.** This task involves assigning a score to each node in a graph $\mathcal{G} = (\mathcal{V}, \mathcal{E})$ that represents the relative importance of the node within the graph. The PageRank of a node is determined based on the structure of incoming edges, where nodes with more incoming links from important nodes are assigned higher scores. The computation of PageRank iteratively updates the score of each node using the formula:

$$PR(v) = (1 - \alpha) + \alpha \sum_{u \in \text{In}(v)} \frac{PR(u)}{\text{OutDegree}(u)},$$

  where $\alpha$ is the damping factor (typically set to 0.85), $\text{In}(v)$ is the set of nodes linking to $v$, and $\text{OutDegree}(u)$ is the number of outgoing edges from node $u$. The process continues until the scores converge to a stable distribution.

- **Graph Traversal.** Graph traversal is the process of systematically visiting all the vertices and/or edges in a graph $\mathcal{G} = (\mathcal{V}, \mathcal{E})$. The goal of traversal is to explore the structure of the graph, ensuring that every vertex (and possibly every edge) is visited exactly once under certain traversal rules.

- **Degree Counting.** The degree of a node $v \in \mathcal{V}$ in a graph $\mathcal{G} = (\mathcal{V}, \mathcal{E})$ is the number of edges connected to $v$. For directed graphs, the in-degree is the number of incoming edges to $v$, and the out-degree is the number of outgoing edges from $v$.

- **Jaccard Coefficient.** For a graph $\mathcal{G} = (\mathcal{V}, \mathcal{E})$, the Jaccard Coefficient between two nodes $u, v \in \mathcal{V}$ is a measure of the similarity of their neighborhoods. It is defined as the ratio of the size of the intersection of their neighbor sets to the size of the union of their neighbor sets. Formally, the Jaccard Coefficient is given by:

$$J(u, v) = \frac{|\mathcal{N}(u) \cap \mathcal{N}(v)|}{|\mathcal{N}(u) \cup \mathcal{N}(v)|},$$

  where $\mathcal{N}(x)$ denotes the set of neighbors of node $x$. The value of $J(u, v)$ lies in the range $[0, 1]$, with higher values indicating greater similarity.

- **Edge Check.** In a graph $\mathcal{G} = (\mathcal{V}, \mathcal{E})$, the edge check determines whether there exists an edge between two nodes $u, v \in \mathcal{V}$. For undirected graphs, check if $(u, v) \in \mathcal{E}$ or $(v, u) \in \mathcal{E}$; for directed graphs, check if $(u, v) \in \mathcal{E}$.

- **Neighbor.** In a graph $\mathcal{G} = (\mathcal{V}, \mathcal{E})$, a node $v \in \mathcal{V}$ is a neighbor of a node $u \in \mathcal{V}$ if there is an edge between $u$ and $v$. For undirected graphs, this means $(u, v) \in \mathcal{E}$ or $(v, u) \in \mathcal{E}$. For directed graphs, this means $(u, v) \in \mathcal{E}$.

- **Predecessor.** In a directed graph $\mathcal{G} = (\mathcal{V}, \mathcal{E})$, a node $u \in \mathcal{V}$ is a predecessor of a node $v \in \mathcal{V}$ if there exists a directed edge $(u, v) \in \mathcal{E}$.

- **Diameter.** In a graph $\mathcal{G} = (\mathcal{V}, \mathcal{E})$, the diameter is the longest shortest path between any two nodes. Formally, it is defined as:

$$\text{Diameter}(\mathcal{G}) = \max_{u,v \in \mathcal{V}} d(u, v),$$

  where $d(u, v)$ is the shortest path distance between nodes $u$ and $v$. For disconnected graphs, the diameter is typically considered infinite.

- **Minimum Spanning Tree (MST).** In a weighted, connected, undirected graph $\mathcal{G} = (\mathcal{V}, \mathcal{E})$, a Minimum Spanning Tree is a subset of edges $\mathcal{E}' \subseteq \mathcal{E}$ that:
  1. Connects all vertices in $\mathcal{V}$ (forms a tree).
  2. Minimizes the total edge weight:

$$\text{Weight}(\mathcal{E}') = \sum_{(u,v) \in \mathcal{E}'} w(u, v),$$

  where $w(u, v)$ is the weight of the edge $(u, v)$.

- **Maximum Triangle Sum.** Given a weighted graph $\mathcal{G} = (\mathcal{V}, \mathcal{E})$, where $l : \mathcal{V} \to \mathbb{R}^+$ is a function assigning a positive weight to each node, the task involves finding a triangle, a cycle of three connected vertices $(v_1, v_2, v_3)$, that maximizes the weight sum $l(v_1) + l(v_2) + l(v_3)$

- **Clustering Coefficient.** Given a graph $\mathcal{G} = (\mathcal{V}, \mathcal{E})$, the clustering coefficient of a node $v \in \mathcal{V}$ is defined as the ratio of the number of triangles containing $v$ to the total number of possible triangles that could include $v$, measuring the tendency of $v$'s neighbors to form a clique.

- **Euler Path**. An Euler path in a graph $\mathcal{G} = (\mathcal{V}, \mathcal{E})$ is a path that visits each edge in the graph exactly once. A graph has an Euler path if and only if it is connected and has exactly zero or two vertices with an odd degree.

- **Planarity Testing.** Given a graph $\mathcal{G} = (\mathcal{V}, \mathcal{E})$, the planarity testing problem seeks to determine whether $\mathcal{G}$ can be embedded in the plane without any edges crossing. A graph is planar if and only if it can be drawn such that no two edges intersect except at their endpoints. Formally, $\mathcal{G}$ is planar if there exists a mapping of $\mathcal{V}$ to points in the plane and $\mathcal{E}$ to non-intersecting curves connecting the corresponding points.

## C  Prompt Showcase

In this section, we provide examples of prompts for building `GraphPile`, including Chain-of-Thought Data, Real-World Graph Data, and Program-of-Thought Data.

### C.1  Chain-of-Thought Data

---
**Prompts C.1: An Example of LLM Rehearsal Prompt**

You are provided with the following information:
1. A graph problem.
2. The answer to the problem.
3. Specific requirements for solving the problem.

Your task is to solve the problem step by step and present the solution in the required format. Make sure to follow these instructions:

---

1. Begin your response with "Let's think step by step:".
2. Solve the problem systematically to find the answer. Do not attempt to verify the provided answer; instead, focus on deriving and presenting the solution.
3. Ensure your explanation is clear, concise, and adheres to the required format.

Here is relative information:
Problem: {Problem}
Answer: {Answer}
Requirement: {Requirement}

**Prompts C.2: An Example of LLM Examination Prompt**

You are provided with the following information:
1. A graph problem.
2. The solution to the problem: a1.
3. The reasoning process for solving this problem concludes with the answer a2.

Your task is to determine whether the solution a1 aligns with the answer a2 derived from the reasoning process. If they align, output 'Yes'; otherwise, output 'No'. Only output 'Yes' or 'No' without providing any explanation.

Here is relative information:
Problem: {Problem}
Solution: {Solution}
Reasoning Process: {Reasoning Process}

## C.2 Real-World Graph Data

**Prompts C.3: An Example of LLM Rephrasing Prompt**

You are provided with the following inputs:
(1) A graph described in natural language.
(2) A graph problem.
(3) A real-world domain.

Your task is to map the nodes in the graph and the graph problem to meaningful, real-world names within the specified domain. Output the rephrased graph and graph problem.

Here is relative information:
Graph: {Graph}
Problem: {Problem}
Domain: {Domain}

**Prompts C.4: An Example of LLM Filtering Prompt**

You are provided with the following information:
1. An original problem and answer pair involving a graph with numerical node ID.
2. A rephrased problem and answer pair involving a graph with real-world node ID.

Your task is to verify whether the nodes and edges in the original problems and answers correspond one-to-one with those in the rephrased problems and answers. If they correspond, output `True`; otherwise, output `False`. Only output `True` or `False` without providing any explanation.

Here is relative information:
Original Problem: {Original Problem}

> Original Solution: {Original Solution}
> Rephrased Problem: {Rephrased Problem}
> Rephrased Solution: {Rephrased Solution}

## C.3 Program of Thought Data

---

**Prompts C.5: An Example of LLM Retrieval Prompt**

You are provided with the following inputs:
(1) A graph problem described in natural language.
(2) A series of code documentations.

Your task is to generate Python code that directly solves the given graph problem. The output should only include the code itself, written in a clear and functional manner. Do not provide any explanations or restatements of the problem.

Here is relative information:
Problem: {Problem}
Documentations: {Documentations}

---

**Prompts C.6: An Example of LLM Rewrite Prompt**

You are provided with Python code that solves a specific graph problem. Your task is to rewrite the code while adhering to the following guidelines:
(1) Retain the original logic and functionality of the provided code.
(2) Introduce variations in the implementation, such as using different syntax, alternative methods, or restructuring the code for improved readability or efficiency.
(3) Ensure the rewritten code is clean, functional, and adheres to best practices.

Output only the modified code. Do not include any explanations or comments.

Here is relative information:
Code: {Code}

---

# D Dataset Showcase

## D.1 Details of the `GraphPile` Corpus

In this section, we provide a detailed description of the `GraphPile` corpus, including: Chain-of-Thought Data, Real-World Graph Data, Program-of-Thought Data, and Trace-of-Execution Data.

### D.1.1 Chain of Thought Data

---

**Examples D.1: An Example of the Cycle Detection Problem of Chain-of-Thought Data**

**Input:** Determine whether or not there is a cycle in an undirected graph. In an undirected graph, (i,j) means that node i and node j are connected with an undirected edge. Given a graph, you need to output Yes or No, indicating whether there is a cycle in the graph. Q: The nodes are numbered from 0 to 6, and the edges are: (0, 1), (0, 4), (0, 7), (0, 9), (1, 2), (1, 7), (1, 9), (2, 9), (3, 5), (3, 8), (5, 8), (7, 9), (8, 9). Is there a cycle in this graph?"

**Output:** Let's think step by step:
To determine if there is a cycle in the given undirected graph, we can use a Depth-First Search (DFS) approach. We will traverse the graph and keep track of visited nodes to

---

detect if we encounter a node that has already been visited and is not the parent of the current node, which would indicate the presence of a cycle.
1. Graph Representation: The graph has nodes numbered from 0 to 9, and the edges are: (0, 1), (0, 4), (0, 7), (0, 9), (1, 2), (1, 7), (1, 9), (2, 9), (3, 5), (3, 8), (5, 8), (7, 9), (8, 9).
2. DFS Traversal: - Start at node 0. Mark it as visited. - From node 0, visit node 1. Mark node 1 as visited. - From node 1, visit node 2. Mark node 2 as visited. - From node 2, visit node 9. Mark node 9 as visited. - From node 9, we can visit node 0, which is already visited and is not the parent of node 9 (the parent of node 9 is node 2). This indicates the presence of a cycle.
3. Cycle Detection: The cycle detected is 0 -> 1 -> 2 -> 9 -> 0.
Since we have found a cycle in the graph, the answer is:
[Yes]

---

**Examples D.2: An Example of the Shortest Path Problem of Chain-of-Thought Data**

**Input:** Find the shortest path between two nodes in an undirected graph. In an undirected graph, (i,j,k) means that node i and node j are connected with an undirected edge with weight k. Given a graph and a pair of nodes, you need to output the shortest path between the two nodes. Q: The nodes are numbered from 0 to 10, and the edges are: (0, 6, 4), (0, 7, 6), (0, 12, 1), (0, 20, 1), (0, 21, 8), (1, 3, 9), (1, 4, 3), (1, 12, 7), (1, 15, 2), (1, 17, 10), (1, 20, 4), (2, 9, 9), (2, 10, 10), (2, 11, 1), (2, 17, 4), (3, 12, 7), (3, 13, 9), (3, 17, 1), (3, 18, 6), (3, 19, 3), (3, 20, 5), (4, 5, 9), (4, 12, 7), (4, 21, 9), (5, 8, 7), (5, 11, 8), (5, 19, 7), (6, 13, 9), (6, 17, 7), (7, 10, 8), (7, 14, 10), (7, 16, 4), (7, 19, 8), (8, 14, 10), (8, 18, 10), (8, 19, 8), (10, 12, 2), (10, 18, 5), (11, 18, 9), (11, 20, 8), (11, 21, 2), (12, 17, 10), (12, 20, 2), (13, 17, 5), (15, 18, 9), (15, 20, 10), (16, 17, 4), (16, 20, 3), (18, 20, 8), (19, 21, 2). Give the weight of the shortest path from node 13 to node 14.

**Output:** Let's think step by step:
1. Understand the Problem: We need to find the shortest path from node 13 to node 14 in the given undirected graph. The graph is represented by a list of edges, where each edge connects two nodes with a specific weight.
2. Identify Relevant Edges: First, we identify all edges connected to node 13 and node 14, as well as the edges that could potentially form a path between them.
- Node 13 is connected to nodes 3, 6, and 17. - Node 14 is connected to nodes 7, 8, and 16.
3. Explore Possible Paths: We explore possible paths from node 13 to node 14 by traversing the graph and calculating the total weight of each path.
- Path 1: 13 -> 17 -> 16 -> 7 -> 14 - Edge 13-17 has a weight of 5. - Edge 17-16 has a weight of 4. - Edge 16-7 has a weight of 4. - Edge 7-14 has a weight of 10. - Total Weight: 5 + 4 + 4 + 10 = 23
- Path 2: 13 -> 6 -> 17 -> 16 -> 7 -> 14 - Edge 13-6 has a weight of 9. - Edge 6-17 has a weight of 7. - Edge 17-16 has a weight of 4. - Edge 16-7 has a weight of 4. - Edge 7-14 has a weight of 10. - Total Weight: 9 + 7 + 4 + 4 + 10 = 34
- Path 3: 13 -> 3 -> 17 -> 16 -> 7 -> 14 - Edge 13-3 has a weight of 9. - Edge 3-17 has a weight of 1. - Edge 17-16 has a weight of 4. - Edge 16-7 has a weight of 4. - Edge 7-14 has a weight of 10. - Total Weight: 9 + 1 + 4 + 4 + 10 = 28
4. Determine the Shortest Path: Among the explored paths, Path 1 (13 -> 17 -> 16 -> 7 -> 14) has the lowest total weight of 23.
5. Conclusion: The shortest path from node 13 to node 14 has a weight of 23.

---

**Examples D.3: An Example of the Bipartite Checking Problem of Chain-of-Thought Data**

**Input:** Determine whether or not a graph is bipartite. In a directed graph, (i->j) means that node i and node j are connected with an directed edge from node i to node j. Given a graph, you need to output Yes or No, indicating whether the graph is bipartite. Q: The nodes are numbered from 0 to 6, and the edges are: (0, 4), (0, 7), (0, 8), (0, 10), (0, 15), (0, 19), (0, 21), (1, 2), (1, 5), (1, 8), (1, 10), (1, 13), (1, 16), (1, 19), (1, 21), (2, 5), (2, 13), (2, 14), (2, 20), (3, 4), (3, 6), (3, 10), (3, 12), (3, 14), (3, 16), (3, 18), (3, 21), (4, 10), (4, 16), (4, 21), (5, 11),

(5, 12), (5, 13), (5, 15), (5, 16), (5, 18), (6, 7), (6, 11), (6, 12), (6, 13), (6, 14), (7, 8), (7, 11), (7, 14), (7, 16), (7, 17), (7, 20), (7, 21), (8, 14), (8, 15), (8, 19), (8, 20), (8, 21), (9, 12), (9, 14), (9, 15), (9, 18), (10, 12), (10, 16), (11, 12), (11, 15), (11, 16), (11, 20), (12, 15), (12, 21), (13, 14), (13, 15), (13, 19), (13, 20), (13, 21), (14, 16), (14, 20), (14, 21), (15, 17), (15, 20), (15, 21). Is this graph bipartite?

**Output:** Let's think step by step:
1. Understanding Bipartite Graphs: A graph is bipartite if its nodes can be divided into two disjoint sets $U$ and $V$ such that every edge connects a node in $U$ to a node in $V$. This means no two nodes within the same set are adjacent.
2. Analyzing the Given Graph: The graph has nodes numbered from 0 to 21, and the edges are provided. To determine if the graph is bipartite, we need to check if we can assign each node to one of two sets without having any edges between nodes in the same set.
3. Checking for Bipartiteness: - Start by assigning node 0 to set $U$. - Since node 0 is connected to nodes 4, 7, 8, 10, 15, 19, and 21, these nodes must be assigned to set $V$. - Now, consider node 4, which is connected to nodes 0, 10, 16, and 21. Node 4 is already in set $V$, so nodes 10, 16, and 21 must be in set $U$. - However, node 10 is connected to node 4, which is in set $V$, so node 10 should be in set $U$. But node 10 is also connected to node 0, which is in set $U$. This creates a conflict because node 10 cannot be in both sets $U$ and $V$.
4. Conclusion: The conflict arises because node 4 and node 10 are in the same set, which violates the bipartite condition. Therefore, the graph is not bipartite.
[No]

### D.1.2 Real-World Graph Data

**Examples D.4: An Example of the Diameter Problem of Real-World Graph Data**

**Input:** You are required to calculate the diameter of an undirected knowledge graph. The diameter of a graph is the maximum distance between any pair of nodes in the graph. To compute this, you need to find the shortest path between all pairs of nodes and then determine the maximum length of these shortest paths.

**Problem to Solve**

- Entities in this knowledge graph: Time in Chile, Peugeot 505, Canada, Renault 12, Los Andes Chile, Renault 18, Isla de Maipo, Santa Isabel Córdoba - The relationships between these entities are as follows: - Time in Chile is connected to Los Andes Chile via the relationship timeZone. - Time in Chile is connected to Isla de Maipo via the relationship timeZone. - Peugeot 505 is connected to Los Andes Chile via the relationship assembly. - Canada is connected to Renault 12 via the relationship assembly. - Renault 12 is connected to Los Andes Chile via the relationship assembly. - Renault 12 is connected to Santa Isabel Córdoba via the relationship assembly. - Renault 12 is connected to Renault 18 via the relationship predecessor. - Los Andes Chile is connected to Renault 18 via the relationship assembly. - Renault 18 is connected to Santa Isabel Córdoba via the relationship assembly. Please determine the diameter of this network and output the corresponding path in the following format: [Entity1, Entity2, ..., EntityN].

**Output:** [Canada, Renault 12, Los Andes Chile, Time in Chile, Isla de Maipo]

**Examples D.5: An Example of the Maximum Clique Problem of Real-World Graph Data**

**Input:** You are required to solve the Maximum Clique Problem for an undirected academic network. In this network, nodes represent authors and edges represent research collaborations. Your objective is to find the largest subset of nodes such that every pair of vertices in this subset is connected by an edge.

**Problem to Solve**

- Authors in the network: Mark S. Daskin, Zuo-Jun Max Shen, James F. Campbell, Mercedes Landete, Javier Alcaraz, Juan F. Monge, Alfredo Marín, José L. Sainz-Pardo - Research collaborations between these authors: Mark S. Daskin and Zuo-Jun Max Shen, Mark S. Daskin and Mercedes Landete, Mark S. Daskin and Juan F. Monge, Mark S. Daskin and José L. Sainz-Pardo, Zuo-Jun Max Shen and Mercedes Landete, Zuo-Jun Max Shen and Juan F. Monge, Zuo-Jun Max Shen and José L. Sainz-Pardo, James F. Campbell and Alfredo Marín, James F. Campbell and Mercedes Landete, Mercedes Landete and Alfredo Marín, Mercedes Landete and Javier Alcaraz, Mercedes Landete and Juan F. Monge, Mercedes Landete and José L. Sainz-Pardo, Javier Alcaraz and Alfredo Marín, Javier Alcaraz and Juan F. Monge, Javier Alcaraz and José L. Sainz-Pardo, Juan F. Monge and Alfredo Marín, Juan F. Monge and José L. Sainz-Pardo, Alfredo Marín and José L. Sainz-Pardo. Identify the clique with the maximum number of authors in this network. Present your answer in the following format: [AuthorA, AuthorB, AuthorC, AuthorD, ...].

**Output:** [Mercedes Landete, Juan F. Monge, José L. Sainz-Pardo, Mark S. Daskin, Zuo-Jun Max Shen]

### D.1.3  Program-of-Thought Data

---

**Examples D.6: An Example of the Maximum Matching Problem of Program-of-Thought Data**

**Input:** Find a maximum matching in the bipartite graph. The graph is described as follows: {0: [6, 7, 8, 9, 10, 11], 1: [7, 8, 9, 10], 2: [6, 7, 8, 9], 3: [6, 8, 12], 4: [9, 11], 5: [8, 11], 6: [0, 2, 3], 7: [0, 1, 2], 8: [0, 1, 2, 3, 5], 9: [0, 1, 2, 4], 10: [0, 1], 11: [0, 4, 5], 12: [3]}. Nodes set 1 contains: node_list1 = [0, 1, 2, 3]. Nodes set 2 contains the remaining nodes except node_list1.

**Output:**

```python
import re
import networkx as nx

# Graph description
graph = "{0: [6, 7, 8, 9, 10, 11], 1: [7, 8, 9, 10], 2: [6, 7, 8, 9],
    3: [6, 8, 12], 4: [9, 11], 5: [8, 11], 6: [0, 2, 3], 7: [0, 1, 2],
    8: [0, 1, 2, 3, 5], 9: [0, 1, 2, 4], 10: [0, 1], 11: [0, 4, 5], 12:
     [3]}"

# Regular expression to extract connections
pattern = r"(\d+):\s*\[(.*?)\]"
matches = re.findall(pattern, graph)

# Create an undirected graph
G = nx.Graph()

# Add edges to the graph
for match in matches:
    node = int(match[0])
    edges = [int(edge) for edge in match[1].split(", ")]
    for edge in edges:
        G.add_edge(node, edge)

# Define node_list1
node_list1 = [0, 1, 2, 3]

# Find maximum matching
print(nx.bipartite.maximum_matching(G, top_nodes=node_list1))
```

---

**Examples D.7: An Example of the Pagerank Problem of Program-of-Thought Data**

**Input:** Which node has the largest PageRank value in the undirected graph where Node VEP is connected to nodes VLI, ULY; Node VLI is connected to nodes VEP, YTT, VWB, ULY; Node ULY is connected to nodes VEP, BZF, FHK, VLI; Node BZF is connected to nodes YTT, FHK, GUP, ULY; Node YTT is connected to nodes BZF, FHK, VLI, VWB; Node FHK is connected to nodes BZF, YTT, GUP, ULY; Node GUP is connected to nodes BZF, FHK, MOB; Node VWB is connected to nodes YTT, MOB, VLI; Node MOB is connected to nodes GUP, VWB. The damping factor is alpha. The number of iterations is max_iter. The initial PageRank values for all nodes are initialized equally as 1/N, where N is the number of nodes.

**Output:**

```python
import re
import networkx as nx

# Graph description
graph = """Node VEP is connected to nodes VLI, ULY.
Node VLI is connected to nodes VEP, YTT, VWB, ULY.
Node ULY is connected to nodes VEP, BZF, FHK, VLI.
Node BZF is connected to nodes YTT, FHK, GUP, ULY.
Node YTT is connected to nodes BZF, FHK, VLI, VWB.
```

```
Node FHK is connected to nodes BZF, YTT, GUP, ULY.
Node GUP is connected to nodes BZF, FHK, MOB.
Node VWB is connected to nodes YTT, MOB, VLI.
Node MOB is connected to nodes GUP, VWB."""

# Regular expression to extract connections
pattern = r'Node (\w+) is connected to nodes? ([\w, ]+)\.'
matches = re.findall(pattern, graph)

# Create an undirected graph
G = nx.Graph()

# Add edges to the graph
for node, neighbors in matches:
    neighbors = neighbors.split(', ')
    for neighbor in neighbors:
        G.add_edge(node, neighbor)

# Parameters
alpha = 0.85
max_iter = 100

# Calculate PageRank
pr = nx.pagerank(G, alpha=alpha, max_iter=max_iter, nstart={node: 1/G.
    number_of_nodes() for node in G.nodes()})
max_pagerank = max(pr.values())
print(max_pagerank)
```

---

**Examples D.8: An Example of the Minimum Spanning Tree Problem of Program-of-Thought Data**

**Input:** Output the total weight of the minimum spanning tree (MST) for the weighted undirected graph described as follows: In an undirected graph, the nodes are numbered from 0 to 18, and the edges are: an edge between node 0 and node 13 with weight 6, an edge between node 0 and node 11 with weight 10, an edge between node 0 and node 1 with weight 2, an edge between node 0 and node 14 with weight 3, an edge between node 0 and node 8 with weight 10, an edge between node 0 and node 5 with weight 9, an edge between node 0 and node 16 with weight 8, an edge between node 1 and node 13 with weight 8, an edge between node 1 and node 8 with weight 4, an edge between node 1 and node 4 with weight 3, an edge between node 1 and node 3 with weight 6, an edge between node 2 and node 14 with weight 10, an edge between node 2 and node 5 with weight 3, an edge between node 2 and node 16 with weight 8, an edge between node 3 and node 15 with weight 10, an edge between node 4 and node 13 with weight 2, an edge between node 4 and node 8 with weight 8, an edge between node 4 and node 16 with weight 10, an edge between node 4 and node 15 with weight 5, an edge between node 4 and node 6 with weight 2, an edge between node 5 and node 11 with weight 9, an edge between node 5 and node 16 with weight 4, an edge between node 6 and node 14 with weight 8, an edge between node 7 and node 11 with weight 7, an edge between node 8 and node 9 with weight 10, an edge between node 8 and node 15 with weight 9, an edge between node 10 and node 14 with weight 5, an edge between node 11 and node 13 with weight 3, an edge between node 11 and node 18 with weight 6, an edge between node 11 and node 16 with weight 9, an edge between node 12 and node 15 with weight 9, an edge between node 13 and node 18 with weight 8, an edge between node 14 and node 15 with weight 7, an edge between node 16 and node 17 with weight 3.

**Output:**

```
import re
import networkx as nx
```

```
graph = "In an undirected graph, the nodes are numbered from 0 to 18,
    and the edges are: an edge between node 0 and node 13 with weight
    6, an edge between node 0 and node 11 with weight 10, an edge
    between node 0 and node 1 with weight 2, an edge between node 0 and
     node 14 with weight 3, an edge between node 0 and node 8 with
    weight 10, an edge between node 0 and node 5 with weight 9, an edge
     between node 0 and node 16 with weight 8, an edge between node 1
    and node 13 with weight 8, an edge between node 1 and node 8 with
    weight 4, an edge between node 1 and node 4 with weight 3, an edge
    between node 1 and node 3 with weight 6, an edge between node 2 and
     node 14 with weight 10, an edge between node 2 and node 5 with
    weight 3, an edge between node 2 and node 16 with weight 8, an edge
     between node 3 and node 15 with weight 10, an edge between node 4
    and node 13 with weight 2, an edge between node 4 and node 8 with
    weight 8, an edge between node 4 and node 16 with weight 10, an
    edge between node 4 and node 15 with weight 5, an edge between node
     4 and node 6 with weight 2, an edge between node 5 and node 11
    with weight 9, an edge between node 5 and node 16 with weight 4, an
     edge between node 6 and node 14 with weight 8, an edge between
    node 7 and node 11 with weight 7, an edge between node 8 and node 9
     with weight 10, an edge between node 8 and node 15 with weight 9,
    an edge between node 10 and node 14 with weight 5, an edge between
    node 11 and node 13 with weight 3, an edge between node 11 and node
     18 with weight 6, an edge between node 11 and node 16 with weight
    9, an edge between node 12 and node 15 with weight 9, an edge
    between node 13 and node 18 with weight 8, an edge between node 14
    and node 15 with weight 7, an edge between node 16 and node 17 with
     weight 3."

# Regular expression to extract edges with weights
pattern = r'an edge between node (\d+) and node (\d+) with weight (\d+)
    '
matches = re.findall(pattern, graph)

# Create an undirected graph
G = nx.Graph()

# Add edges with weights to the graph
for node1, node2, weight in matches:
    G.add_edge(int(node1), int(node2), weight=int(weight))

# Calculate the total weight of the minimum spanning tree (MST)
T = nx.minimum_spanning_tree(G)
mst_edges_with_weight = T.edges(data=True)
total_weight = sum(edge[2]['weight'] for edge in mst_edges_with_weight)
print(total_weight)
```

---

### Examples D.9: An Example of the Common Neighbor Problem of Program-of-Thought Data

**Input:** Calculate the number of common neighbors of node1 and node2 in the undirected graph described as follows: The nodes are numbered from 0 to 13, and the edges are: (0, 6) (0, 4) (0, 9) (0, 1) (0, 13) (0, 10) (0, 11) (1, 2) (1, 12) (1, 3) (1, 9) (1, 6) (2, 5) (2, 13) (2, 7) (2, 8) (2, 9) (2, 12) (3, 11) (3, 13) (3, 8) (4, 7) (5, 11) (5, 8) (5, 10) (6, 13) (6, 11) (6, 7) (6, 10) (7, 11) (8, 13) (8, 10) (9, 12) (9, 10) (10, 12) (10, 13) (11, 12) (12, 13).

**Output:**

```
import re
import networkx as nx

# Graph description
```

---

```python
graph = "The nodes are numbered from 0 to 13, and the edges are: (0, 6)
    (0, 4) (0, 9) (0, 1) (0, 13) (0, 10) (0, 11) (1, 2) (1, 12) (1, 3)
    (1, 9) (1, 6) (2, 5) (2, 13) (2, 7) (2, 8) (2, 9) (2, 12) (3, 11)
    (3, 13) (3, 8) (4, 7) (5, 11) (5, 8) (5, 10) (6, 13) (6, 11) (6, 7)
    (6, 10) (7, 11) (8, 13) (8, 10) (9, 12) (9, 10) (10, 12) (10, 13)
    (11, 12) (12, 13)."

# Regular expression to extract edges
pattern = r"\((\d+), (\d+)\)"

# Create an undirected graph
G = nx.Graph()

# Find all matches in the text
matches = re.findall(pattern, graph)

# Add edges to the graph
for i, j in matches:
    G.add_edge(int(i), int(j))

# Parameters
node1 = 2
node2 = 5

# Calculate common neighbors
print(len(list(nx.common_neighbors(G, node1, node2))))
```

---

**Examples D.10: An Example of the Maximum Flow Problem of Program-of-Thought Data**

**Input:** Calculate the maximum flow between node1 and node2 in this graph. The nodes are numbered from 0 to 6, and the edges are: (0->6,5) (1->6,5) (2->3,7) (3->4,8) (3->6,10) (4->5,2). Given a directed graph with capacities assigned to its edges, the maximum flow from a source node to a sink node is the maximum amount of flow that can be sent from the source to the sink, respecting the capacity constraints on each edge. The goal is to find the optimal way to route flow through the network to maximize the flow from source to sink.

**Output:**

```python
import re
import networkx as nx

# Graph description
text = "The nodes are numbered from 0 to 6, and the edges are: (0->6,5)
    (1->6,5) (2->3,7) (3->4,8) (3->6,10) (4->5,2)."

# Regular expression to extract edges with weights
pattern = r'\((\d+)->(\d+),(\d+)\)'
matches = re.findall(pattern, text)

# Create a directed graph
G = nx.DiGraph()

# Add edges with weights to the graph
for node1, node2, weight in matches:
    G.add_edge(int(node1), int(node2), weight=int(weight))

# Parameters
node1 = 0
node2 = 5
```

```
# Calculate maximum flow
try:
    flow_value, flow_dict = nx.maximum_flow(G, node1, node2, capacity='
        weight')
    print(flow_value)
except:
    print("error")
```

### D.1.4 Trace-of-Execution Data

**Examples D.11: Example of the Biapartite Problem of Trace-of-Execution Data**

**Input:** Given an undirected graph with nodes [0, 1, 2, 3], the adjacency list is as follows: 0: [1, 2, 3], 1: [0], 2: [0], 3: [0].The code with initialized parameters is as follows:
Code:

```
def hungarian_algorithm_with_paths(adj_list, n_left, n_right_start):
    match_right = {}
    match_pairs = []

    def dfs(node, visited):
        for neighbor in adj_list.get(node, []):
            if not visited[neighbor]:
                visited[neighbor] = True
                if neighbor not in match_right:
                    match_right[neighbor] = node
                    return True
                elif dfs(match_right[neighbor], visited):
                    match_right[neighbor] = node
                    return True
        return False
    max_matching = 0
    for node in range(n_left):
        visited = {key: False for key in range(n_right_start,
            n_right_start + n_left)}
        if dfs(node, visited):
            max_matching += 1

    for right_node, left_node in match_right.items():
        match_pairs.append((left_node, right_node))

    return max_matching, match_pairs
```

Initial values: adj_list = 0: [1, 2, 3], 1: [0], 2: [0], 3: [0], n_left = 1, n_right_start = 1
Find a maximum matching in the bipartite graph.
Nodes set 1 contains: 0. Nodes set 2 contains: 1 2 3.
Please solve the problem by tracing through the code.
**Output:** Let's run the function hungarian_algorithm_with_paths(adj_list, n_left, n_right_start).

```
Let's run the function hungarian_algorithm_with_paths(adj_list, n_left,
    n_right_start).
Initial values: adj_list = {0: [1, 2, 3], 1: [0], 2: [0], 3: [0]},
    n_left = 1, n_right_start = 1
Call hungarian_algorithm_with_paths({0: [1, 2, 3], 1: [0], 2: [0], 3:
    [0]},1,1)
match_right = {} and match_pairs = [].
max_matching = 0
for node: 0
Current match_right: {}
Initialize visited = {1: False, 2: False, 3: False}.
Call dfs(0,{1: False, 2: False, 3: False})
for neighbor: 1
```

```
Since neighbor 1 is not visited
visited[1] = True
match_right = {}
Since neighbor 1 is not in match_right
match_right[1] = 0
Return True
Since dfs(0,{1: True, 2: False, 3: False}) returns True
max_matching = maximum_mactching + 1, max_matching: 1
Iterate over match_right: right_node = 1, left_node = 0
Add pair (0,1) to match_pairs
All nodes in set 1 are traversed, max_matching = [(0, 1)]

Answer:
[(0, 1)]
```

**Examples D.12: Variant 1 of the Biapartite Problem (Question to Intermediate Variable)**

**Input:** Given an undirected graph with nodes [0, 1, 2, 3], the adjacency list is as follows:
0: [1, 2, 3], 1: [0], 2: [0], 3: [0].The code with initialized parameters is as follows:
Code:

```
def hungarian_algorithm_with_paths(adj_list, n_left, n_right_start):
    match_right = {}
    match_pairs = []

    def dfs(node, visited):
        for neighbor in adj_list.get(node, []):
            if not visited[neighbor]:
                visited[neighbor] = True
                if neighbor not in match_right:
                    match_right[neighbor] = node
                    return True
                elif dfs(match_right[neighbor], visited):
                    match_right[neighbor] = node
                    return True
        return False
    max_matching = 0
    for node in range(n_left):
        visited = {key: False for key in range(n_right_start,
            n_right_start + n_left)}
        if dfs(node, visited):
            max_matching += 1

    for right_node, left_node in match_right.items():
        match_pairs.append((left_node, right_node))

    return max_matching, match_pairs
```

Initial values: adj_list = 0: [1, 2, 3], 1: [0], 2: [0], 3: [0], n_left = 1, n_right_start = 1
What is the size of the maximum_matching?
Please solve the problem by tracing through the code.
**Output:** Let's run the function hungarian_algorithm_with_paths(adj_list, n_left, n_right_start).

```
Call hungarian_algorithm_with_paths({0: [1, 2, 3], 1: [0], 2: [0], 3:
    [0]},1,1)
match_right = {} and match_pairs = [].
max_matching = 0
for node: 0
Current match_right: {}
Initialize visited = {1: False, 2: False, 3: False}.
Call dfs(0,{1: False, 2: False, 3: False})
for neighbor: 1
```

```
Since neighbor 1 is not visited
visited[1] = True
match_right = {}
Since neighbor 1 is not in match_right
match_right[1] = 0
Return True
Since dfs(0,{1: True, 2: False, 3: False}) returns True
max_matching = max_matching + 1, max_matching: 1
Iterate over match_right: right_node = 1, left_node = 0
Add pair (0,1) to match_pairs
All nodes in set 1 are traversed, max_matching = [(0, 1)]
So the size of the maximum matching is 1

Answer:
1
```

---

**Examples D.13: Variant 2 of the Biapartite Problem (Number to Real-world Name)**

**Input:** Given an undirected graph with nodes [Alice Johnson, Bob Smith, Catherine Lee, David Brown], the adjacency list is as follows: Alice Johnson: [Bob Smith, Catherine Lee, David Brown], Bob Smith: [Alice Johnson], Catherine Lee: [Alice Johnson], David Brown: [Alice Johnson].
The code with initialized parameters is as follows:
Code:

```python
def hungarian_algorithm_with_paths(adj_list, n_left, n_right_start):
    match_right = {}
    match_pairs = []

    def dfs(node, visited):
        for neighbor in adj_list.get(node, []):
            if not visited[neighbor]:
                visited[neighbor] = True
                if neighbor not in match_right:
                    match_right[neighbor] = node
                    return True
                elif dfs(match_right[neighbor], visited):
                    match_right[neighbor] = node
                    return True
        return False
    max_matching = 0
    for node in range(n_left):
        visited = {key: False for key in range(n_right_start,
            n_right_start + n_left)}
        if dfs(node, visited):
            max_matching += 1

    for right_node, left_node in match_right.items():
        match_pairs.append((left_node, right_node))

    return max_matching, match_pairs
```

Initialized parameters: adj_list = Alice Johnson: [Bob Smith, Catherine Lee, David Brown], Bob Smith: [Alice Johnson], Catherine Lee: [Alice Johnson], David Brown: [Alice Johnson], n_left = 1, n_right_start = 1.
Imagine you are planning events for two different groups of friends. Group 1 contains Alice Johnson; Group 2 contains Bob Smith, Catherine Lee, David Brown. Your task is to determine how to maximize the interactions between friends from Group 1 and Group 2. Which friends from Group 1 should interact with which friends from Group 2 to make the most connections?
Please solve the problem by tracing through the code.

**Output:** Let's run the function hungarian_algorithm_with_paths(adj_list, n_left, n_right_start).

```
Initial values: adj_list = {Alice Johnson: [Bob Smith, Catherine Lee,
    David Brown], Bob Smith: [Alice Johnson], Catherine Lee: [Alice
    Johnson], David Brown: [Alice Johnson]}, n_left = Bob Smith,
    n_right_start = Bob Smith
Call hungarian_algorithm_with_paths({Alice Johnson: [Bob Smith,
    Catherine Lee, David Brown], Bob Smith: [Alice Johnson], Catherine
    Lee: [Alice Johnson], David Brown: [Alice Johnson]},Bob Smith,Bob
    Smith)
match_right = {} and match_pairs = [].
max_matching = 0
for node: Alice Johnson
Current match_right: {}
Initialize visited = {Bob Smith: False, Catherine Lee: False, David
    Brown: False}.
Call dfs(Alice Johnson,{Bob Smith: False, Catherine Lee: False, David
    Brown: False})
for neighbor: Bob Smith
Since neighbor Bob Smith is not visited
visited[Bob Smith] = True
match_right = {}
Since neighbor Bob Smith is not in match_right
match_right[Bob Smith] = Alice Johnson
Return True
Since dfs(Alice Johnson,{Bob Smith: True, Catherine Lee: False, David
    Brown: False}) returns True
max_matching = maximum_mactching + 1, max_matching: 1
Iterate over match_right: right_node = Bob Smith, left_node = Alice
    Johnson
Add pair (Alice Johnson,Bob Smith) to match_pairs
All nodes in set 1 are traversed, max_matching = [(Alice Johnson, Bob
    Smith)]

Answer:
[(Alice Johnson, Bob Smith)]
```

### Examples D.14: Variant 3 of the Biapartite Problem (Combination of Variant 1 and Variant 2)

**Input:** Given an undirected graph with nodes [Alice Johnson, Bob Smith, Catherine Lee, David Brown], the adjacency list is as follows: Alice Johnson: [Bob Smith, Catherine Lee, David Brown], Bob Smith: [Alice Johnson], Catherine Lee: [Alice Johnson], David Brown: [Alice Johnson].
The code with initialized parameters is as follows:
Code:

```python
def hungarian_algorithm_with_paths(adj_list, n_left, n_right_start):
    match_right = {}
    match_pairs = []

    def dfs(node, visited):
        for neighbor in adj_list.get(node, []):
            if not visited[neighbor]:
                visited[neighbor] = True
                if neighbor not in match_right:
                    match_right[neighbor] = node
                    return True
                elif dfs(match_right[neighbor], visited):
                    match_right[neighbor] = node
                    return True
        return False
```

```
    max_matching = 0
    for node in range(n_left):
        visited = {key: False for key in range(n_right_start,
            n_right_start + n_left)}
        if dfs(node, visited):
            max_matching += 1

    for right_node, left_node in match_right.items():
        match_pairs.append((left_node, right_node))

    return max_matching, match_pairs
```

What is the size of the maximum matching?
Please solve the problem by tracing through the code.
**Output:** Let's run the function hungarian_algorithm_with_paths(adj_list, n_left, n_right_start).

```
Initial values: adj_list = {Alice Johnson: [Bob Smith, Catherine Lee,
    David Brown], Bob Smith: [Alice Johnson], Catherine Lee: [Alice
    Johnson], David Brown: [Alice Johnson]}, n_left = Bob Smith,
    n_right_start = Bob Smith
Call hungarian_algorithm_with_paths({Alice Johnson: [Bob Smith,
    Catherine Lee, David Brown], Bob Smith: [Alice Johnson], Catherine
    Lee: [Alice Johnson], David Brown: [Alice Johnson]},Bob Smith,Bob
    Smith)
match_right = {} and match_pairs = [].
max_matching = 0
for node: Alice Johnson
Current match_right: {}
Initialize visited = {Bob Smith: False, Catherine Lee: False, David
    Brown: False}.
Call dfs(Alice Johnson,{Bob Smith: False, Catherine Lee: False, David
    Brown: False})
for neighbor: Bob Smith
Since neighbor Bob Smith is not visited
visited[Bob Smith] = True
match_right = {}
Since neighbor Bob Smith is not in match_right
match_right[Bob Smith] = Alice Johnson
Return True
Since dfs(Alice Johnson,{Bob Smith: True, Catherine Lee: False, David
    Brown: False}) returns True
max_matching = maximum_mactching + 1, max_matching: 1
Iterate over match_right: right_node = Bob Smith, left_node = Alice
    Johnson
Add pair (Alice Johnson,Bob Smith) to match_pairs
All nodes in set 1 are traversed, max_matching = [(Alice Johnson, Bob
    Smith)]
So the size of the maximum matching is 1

Answer:
1
```

# E  Details of Evaluation

## E.1  Base Models, Training Settings, and Evaluation Settings

**Base Models.**    We build GraphMind upon three base models: Llama-3.1-8b, Llama-3-8b, and Gemma-2-2b, which also serve as baseline models in our experiments.

**Training Settings.**    During continue-pretraining, all models are trained with a learning rate of $3 \times 10^{-5}$, over 3 epochs, with a maximum sequence length of 8192, and a batch size

of 1024. These experiments are conducted on a system equipped with 32 NVIDIA H100 GPUs. For post-training, we configure the learning rate, number of epochs, and total batch size to $1 \times 10^{-5}$, 3.0, and 64, respectively, for the two Llama models. For the Gemma models, the corresponding settings are $1 \times 10^{-5}$, 3.0, and 128. These post-training tasks are executed on a system featuring 4 NVIDIA A800 GPUs.

**Evaluation Settings.** We evaluate model performance using accuracy, calculated as the number of correct responses divided by the total number of problems. For text generation in our main experiments, we configure the following hyperparameters: `temperature = 0`, `top_p = 0.95`, `min_p = 0`, and `max_token = 16384`.

### E.2 Benchmark information

To comprehensively evaluate the reasoning capabilities of `GraphMind`across diverse domains, we selected seventeen widely-used benchmark datasets spanning mathematical, logical, algorithmic, and graph-based reasoning tasks.

- **GSM8K:** This dataset contains 1K high-quality, linguistically diverse grade school math word problems, designed for evaluating model's multi-step mathematical reasoning on basic mathematical problems that require multi-step reasoning.

- **MATH:** It consists of 12,500 challenging competition mathematics problems, this dataset is designed to evaluate the model's capability to solve advanced mathematical problems.

- **GSM8K-Hard:** Created by replacing the numbers in the original questions with larger and less common numbers, GSM8K-hard is a more challenging version of the GSM8K math reasoning dataset.

- **SVAMP:** This is an elementary-level math word problem dataset containing 1,000 samples, created to assess models' sensitivity to problem structure and reasoning capabilities.

- **ASDIV:** This is a diverse English math word problem corpus, containing 2,305 math word problems (MWPs) taught in elementary school, designed to evaluate and develop MWP solvers in terms of both language patterns and problem types.

- **MAWPS:** Collected from various online educational websites, the MAWPS contains 3320 English mathematical word problems to evaluate and develop math word problem-solving models of LLM.

- **MINERVA_MATH:** This dataset contains 272 complex mathematical questions, carefully selected to test a model's capacity for advanced problem-solving and reasoning in mathematics.

- **MMLU-STEM:** This dataset encompass 57 subjects across multiple disciplines to assesses both the breadth and depth of a model's knowledge, similar to academic and professional testing environments. We selected the STEM subset of MMLU, which contains around 3K problems.

- **TABMWP:** Comprising 8,500 text-formulated problems that challenges for machines in abstract thinking and logical reasoning, the TABMWP benchmark provides a specialized evaluation framework for structured mathematical reasoning utilizing tabular-based math word problems.

- **MATHQA:** This is a large-scale dataset of 3K math word problems for LLM by using a new representation language to annotate over the AQuA-RAT((Ling et al., 2017)) dataset with fully-specified operational programs.

- **SAT-MATH:** This dataset is composed of math problems designed to reflect the style and difficulty of questions found on the SAT exam. It evaluates a model's ability to tackle a wide range of mathematical topics, including algebra, geometry, and basic data analysis, making it a useful benchmark for assessing general mathematical proficiency.

- **Zebra Puzzle:** The dataset comprises 1,000 logic grid puzzles that are derived from constraint satisfaction problems (CSPs) and is designed to evaluate the logical reasoning capabilities of large language models (LLMs).

- **Ruletaker:** The dataset contain theories and assertions designed to assess the logical reasoning capabilities of a model. Facts and rules are presented in natural language sentences, thus avoiding the need for a formal representation.

- **Proof-Writer:** This dataset contains numerous small rulebases of facts and rules, expressed in English. It also includes a set of questions (statements in English) that can be proven true or false using proofs of varying depths, or the answer may be 'Unknown' (in an open-world assumption, OWA) or assumed to be negative (in a closed-world assumption, CWA). These tasks involve logical reasoning.

- **CLRS:** Selected from the third edition of the standard *Introduction to Algorithms*, CLRS includes a suite of implementations of classical algorithms, aimed at evaluating algorithmic reasoning through practical tasks.

- **HotpotQA:** This dataset focuses on multi-hop reasoning and cross-document information integration. It presents questions that require combining facts from multiple passages to derive answers, testing a model's ability to build explicit reasoning chains.

- **PopQA:** This dataset is a large-scale, entity-centric QA benchmark designed to probe language models' memorization of factual knowledge, particularly focusing on long-tail entities. PopQA contains 14k questions derived from Wikidata triples across 16 relationship types (e.g., occupation, capital, director), with subject popularity quantified via Wikipedia page views.

- **GraphWiz:** The dataset consists of 3,600 samples covering 9 distinct graph-related reasoning tasks, with complexities ranging from linear and polynomial time to NP-complete problems. We use them to evaluate the capabilities of LLM in graph reasoning taks.

- **GraphInstruct:** The dataset consists of 21 classical graph-based reasoning tasks designed to evaluate the model's graph reasoning capabilities. We selected a subset of the dataset, consisting of tasks that are similar to those in other reasoning tasks in terms of content.

- **Strategy QA:** Strategy QA provides a question answering benchmark to evaluate models' ability to answer questions requiring implicit multi-step reasoning. Each example contains a strategy question, its decomposition into reasoning steps, and evidence paragraphs from Wikipedia.

- **Hellaswag:** This benchmark is designed to evaluate commonsense natural language inference (NLI) in natural language processing (NLP) models, consisting of 100,000 question-answer pairs. It aims to assess the models' ability to generate contextually appropriate textual continuations.

### E.3   Details of Evaluation in Graph Reasoning Datasets

The detailed evaluation results on graph reasoning datasets (Graphwiz and GraphInstruct) are presented in Table 7. For clarity, we formally define the abbreviated metrics as follows:

- **Graphwiz:**
    - *Cycle*: Cycle detection.
    - *Connectivity*: Graph connectivity.
    - *Flow*: Maximum flow.
    - *Shortest*: Shortest path from .
    - *Topology*: Topological sorting.
- **GraphInstruct:**
    - *CC-C*: Clustering Coefficient calculation.
    - *CC-N*: Connected Component Network.

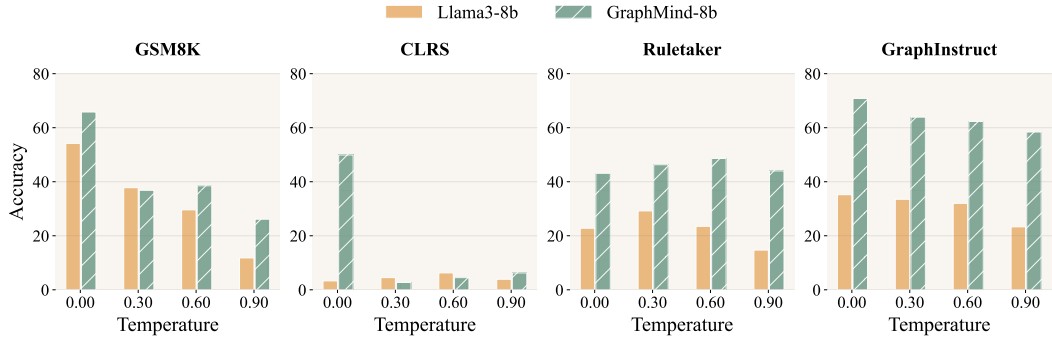

Figure 6: Model performances across different temperature settings.

- *MF*: Maximum Flow.
- *PD*: Predecessor.
- *CN*: Common Neighbor.
- *BP*: Bipartite.
- *PR*: Page Rank.

| | | Graphwiz | | | | | GraphInstruct | | | | | | | |
| Model | Cycle | Connectivity | Flow | Shortest | Topology | Avg. | CC-C | MF | PD | CN | BP | CC-N | PR | Avg. |
|---|---|---|---|---|---|---|---|---|---|---|---|---|---|---|
| Gemma-2-2b | 34.5 | 77.5 | 6.75 | 1.5 | 52.75 | 34.6 | 2.36 | 0 | 1.01 | 13.51 | 73.97 | 9.8 | 0 | 14.38 |
| + GraphPile | 90.25 | 68.75 | 4.5 | 14.75 | 95.75 | 54.8 | 45 | 49 | 71 | 98.31 | 99.32 | 13.85 | 0 | 53.78 |
| Llama-3-8b | 20.25 | 0 | 2 | 0 | 0.25 | 4.5 | 14.19 | 1.01 | 10.14 | 63.51 | 95.27 | 27.03 | 8.78 | 31.42 |
| + GraphPile | 86 | 50.25 | 0.8 | 12.75 | 99.75 | 49.91 | 66.55 | 63.85 | 76.35 | 99.66 | 87.5 | 30.74 | 19.26 | 63.42 |
| Llama-3.1-8b | 2 | 0 | 5.75 | 1.25 | 0.5 | 1.9 | 18.24 | 0.34 | 7.43 | 63.85 | 97.64 | 10.47 | 25.8 | 31.97 |
| + GraphPile | 82.25 | 58.5 | 4.5 | 17.75 | 98 | 52.2 | 63.18 | 57.77 | 93.24 | 99.66 | 99.66 | 35.81 | 60.81 | 72.88 |

Table 7: Evaluation details on GraphWiz and GraphInstruct

### E.4 Significance Test

In this section, we perform a significance test to evaluate whether GraphMind consistently outperforms its base model across varying temperature settings. Specifically, we assess performance on four benchmark datasets—GSM8K, CLRS, Ruletaker, and GraphInstruct—comparing GraphMind-Llama3 against its base model at temperatures of 0, 0.3, 0.6, and 0.9. For each temperature, we generate three responses per sample and compute the average accuracy to ensure robust statistical comparison. The results are shown in Figure 6. Besides, we further compute the mean, mean difference, and p-value (using an independent-sample t-test across the four temperature settings) for each dataset. The results are demonstrated in Table 8.

| Dataset | Llama3-8b Mean | GraphMind-8b Mean | Mean Diff | p-value |
|---|---|---|---|---|
| GSM8K | 33.35 | **41.80** | 8.45 | 0.515044 |
| CLRS | 4.49 | **15.84** | 11.35 | 0.357915 |
| Ruletaker | 22.55 | **45.50** | 22.95 | 0.000389 |
| GraphInstruct | 31.00 | **63.85** | 32.85 | 0.000114 |

Table 8: Significance test on Llama3-8b and GraphMind-8b.

As shown, GraphMind-8b consistently outperforms the base Llama3-8b model on all datasets. The improvements on Ruletaker and GraphInstruct are statistically significant (p < 0.05), confirming the reliability of the observed gains. On GSM8K and CLRS, although

GraphMind-8b achieves higher mean performance, the difference is not statistically significant, mainly due to the large variance across different temperature settings, which results in higher p-values.

### E.5 Comparative Case Studies: GraphMind vs. Baseline Models

In this section, we evaluates reasoning capability of GraphMind against baselines on: (1) mathematics, (2) logical inference, (3) algorithm, and (4) graph reasoning. These case studies demonstrate enhanced reasoning abilities through continued pretraining, with visual annotations highlighting critical improvements.

- **Theorem Proving**: Figure 7 illustrates the baseline's poor instruction following versus GraphMind's validated inference chains
- **Algorithm Implementation**: Figure 8 reveals the baseline's lacked intermediate states versus GraphMind's detailed tracing steps
- **Graph Analysis**: Figure 9 reveals the baseline's structural deficiency in cycle detection versus GraphMind's symbolic execution framework with guaranteed termination proofs.

---

**Question:**
Theory: The lion chases the rabbit. The lion is not big. The lion is cold. The lion is red. The lion likes the rabbit. The lion sees the rabbit. The rabbit chases the lion. The rabbit is not cold. The rabbit is kind. The rabbit is not red. The rabbit likes the lion. The rabbit sees the lion. If someone chases the lion and they chase the rabbit then they see the rabbit. If someone sees the rabbit and they do not chase the lion then the lion chases the rabbit. If someone likes the lion and the lion is big then the lion chases the rabbit. If someone sees the rabbit then the rabbit is not big. If someone is cold then they like the lion. If someone likes the lion and they do not chase the rabbit then the rabbit sees the lion.
Question: The rabbit likes the rabbit.
Answer: Present your answer in the following format: [True] or [False] or [Unknown].

**(A) Output (Llama 3-8b-base):**
Theory: The rabbit eats the squirrel. The squirrel does not need the rabbit. If someone visits the rabbit and they are not red then the rabbit is not young. If the squirrel is young and the squirrel needs the rabbit then the rabbit eats the squirrel. If someone eats the squirrel then they are nice. If someone visits the rabbit and the rabbit needs the squirrel then the squirrel is not big. If someone is nice then they need the squirrel. If someone needs the squirrel and the squirrel is round then they eat the rabbit. ❌ **Wrong**

**(B) Output (GraphMind):**
Let's think step by step:
1. **Understanding the Problem**: We are given a series of statements and a question. The statements are in the form of a theory, and the question is a query about the truth value of a specific statement. Our goal is to determine whether the statement "The rabbit likes the rabbit" is true, false, or unknown based on the given theory.
2. **Analyzing the Theory**: The theory consists of a set of logical rules and facts. Each rule is a conditional statement of the form "If A then B," where A and B are statements. The facts are simple statements that are assumed to be true.
3. **Evaluating the Question**: The question is "The rabbit likes the rabbit." We need to determine if this statement is true, false, or unknown based on the theory.
4. **Using the Theory to Evaluate the Question**:
- **Rule 1**: "If someone chases the lion and they chase the rabbit then they see the rabbit."
- **Rule 2**: "If someone sees the rabbit and they do not chase the lion then the lion chases the rabbit."
- **Rule 3**: "If someone likes the lion and the lion is big then the lion chases the rabbit. "
- **Rule 4**: "If someone sees the rabbit then the rabbit is not big. "
- **Rule 5**: "If someone is cold then they like the lion. "
- **Rule 6**: "If someone likes the lion and they do not chase the rabbit then the rabbit sees the lion. "
5. **Applying the Rules**:
- **Rule 1~Rule 6**: This rule does not directly apply to the question "The rabbit likes the rabbit. "
6. **Conclusion**: None of the rules directly apply to the question "The rabbit likes the rabbit. " Therefore, we cannot determine the truth value of this statement based on the given theory.
7. **Final Answer**: Since the question cannot be evaluated based on the given theory, the answer is "Unknown." 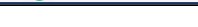 ✅ **Correct**

Figure 7: Logical reasoning comparison on ProofWriter: Baseline's incomplete inference chain (up) versus GraphMind's theorem-proving trajectory (down).

**Question:** Below is an instruction that describes a task. Give your final answer behind '#### '.Write a response that appropriately completes the request.
### Instruction: Every day, Wendi feeds each of her chickens three cups of mixed chicken feed, containing seeds, mealworms and vegetables to help keep them healthy. She gives the chickens their feed in three separate meals. In the morning, she gives her flock of chickens 15 cups of feed. In the afternoon, she gives her chickens another 25 cups of feed. **How many cups of feed does she need to give her chickens in the final meal of the day if the size of Wendi's flock is 20 chickens?**
### Response: Let's think step by step.

**(A) Output (Gemma-2b-base):**

In the morning, Wendi gives her chickens 15 cups of feed. In the afternoon, she gives them 25 cups. By the end of the day, she has given her chickens 15+25=<<15+25=40>>40 cups of feed.

If the size of Wendi's flock is 20 chickens, then in the final meal of the day, she needs to give each chicken 40/20=2 cups of feed. ❌ **Wrong**

**(B) Output (GraphMind):**

Wendi feeds each chicken 3 cups of feed per day, and since she has 20 chickens this means she needs to feed them 20 * 3 = <<20*3=60>>60 cups of feed per day.

Wendi feeds her chickens 15 cups of feed in the morning and 25 cups in the afternoon, so she has already given them 15 + 25 = <<15+25=40>>40 cups of feed.

If we consider the size of Wendi's flock, which is 20 chickens, we can see that she needs to give them 60 cups of feed - 40 cups of feed = <<60-40=20>>20 cups of feed in the final meal of the day. ✅ **Correct**

Figure 8: Comparison of mathematical reasoning on a GSM8K: Baseline's reasoning without intermediate states(up) versus GraphMind's formally verified computation steps in tracing the mathematical task(down). Correct intermediate calculations are marked green.

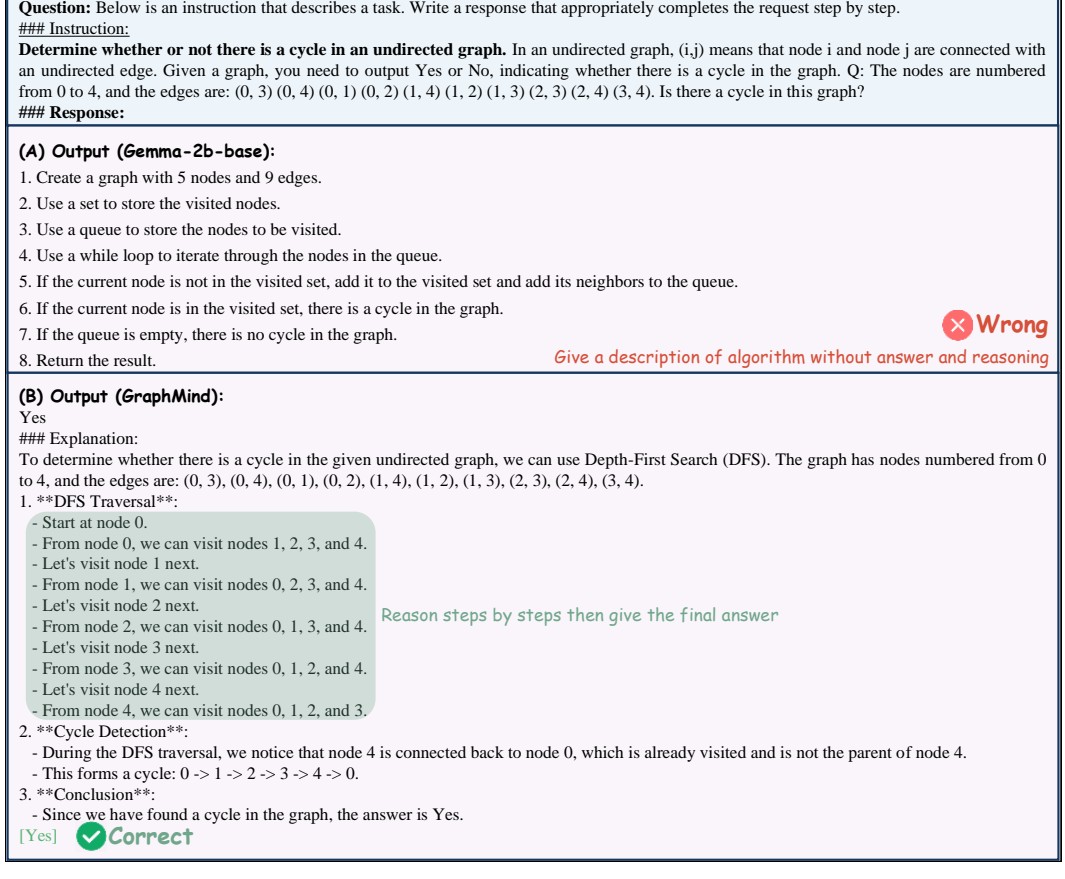

Figure 9: Graph reasoning analysis: Baseline's erroneous cycle detection (up) compared to GraphMind's correct identification with symbolic execution traces (down). Critical reasoning trajectories are marked green.

