# OpenReview forum: "Improving LLMs‘ Generalized Reasoning Abilities by Graph Problems"
_colmweb.org/COLM/2025/Conference — COLM 2025_

### Official Review · Reviewer_bH1x · 2025-04-19

**Rating:** 7
**Confidence:** 4
**Ethics Flag:** 1

**Summary:**

Following Graphwiz and other Graph Problem Reasoning (GPR) works, this paper generates a big graph problem reasoning dataset called GraphPile, which contains 10.9 billion tokens, with different kinds of tasks.

Specifically, it uses real world graphs (and also random graphs), uses different kinds of techniques to generate high quality data:
1. CoT: It uses a program to generate correct answer and then use GPT to translate to CoT language output.
2. PoT: It uses LLM retrievel to find related code for certain problems, and generate programs as output.
3. Trace of Execution: It follows program execution, and tries to predict every step of the program.

I think all these steps are nice way to generate useful information to understand the graph. The authors report that after generating this dataset, one can use it to continue-pretrain the models like LLama and Gemma, and get improvements in both mathematical reasoning and commonsense reasoning.

Quality: I think this is a high quality paper. It works on an important and interesting problem, and provides a series of easy but effective methods.
Clarity: the paper is very easy to follow, with clear figures.
Originality: I am not an expert in this field. But if all these techniques to generate data are original, I think the authors are very creative. So it would be great if the authors can confirm this.
Significance: I think this is a significant result, because it shows how GPR can help improve the power of language models. Moreover, it also provides a nice dataset for the community to use.

**Reasons To Accept:**

I tend to accept this paper for the following reasons:
1. They proposed interesting methods for extracting graph related information from the graph problem.
2. They created a new and big dataset for pretraining, which might be useful for other researchers.
3. They have shown that after using their dataset for continue-pretraining, the model will be more powerful in both math reasoning and commonsense reasoning, which intuitively makes sense to me.
4. The paper is overall well written and easy to follow.

**Reasons To Reject:**

I am not an expert on this GPR field. But if some of the following bullets are true, I will tend to reject this paper:
1. The proposed ways to generate dataset are not new (already proposed by the previous papers)
2. There are already some other similar datasets published, with similar effect (e.g., graphwiz?)
3. The reported improvements on the models (Llama 3&3.1, Gemma 2) are not considered big. (I personally do not have a good judge)
4. There are some other significant problems for the models after continue-pretraining.

---

> ### Author Response · Authors · 2025-06-01
> **Rebuttal by Authors (Part1)**
>
> Thank you for your recognition of our work and your detailed feedback. We greatly appreciate your insightful comments, which have helped us further improve our research. We address your concerns as follows:
>
> * **Q1:** Innovations in GraphPile’s Dataset Generation Methods
> * **Q2:** GraphPile’s Distinct Advantages Over Existing Graph Reasoning Datasets
> * **Q3:** Magnitude and Significance of Model Improvements After Continue-Pretraining on GraphPile
> * **Q4:** Problems Observed After Continue-Pretraining
>
> ---
> ### Q1: **Innovations in GraphPile’s Dataset Generation Methods**
>
> You raise a crucial point-comparing our approach with other data augmentation methods would provide a more comprehensive validation! While previous works have explored various dataset generation methods, our approach introduces **several important innovations** that distinguish GraphPile:
>
> - **Chain-of-Thought (CoT):**
>   Prior works such as GraphWiz [1] used LLMs to generate reasoning paths for graph problems, but **did not verify the correctness of these paths**. In contrast, our work is the first in the LLM graph reasoning domain to propose a **program-guided approach**. Specifically, we write dedicated programs to generate diverse solution paths, ensuring their correctness since they are produced and checked by code execution. We then use LLMs as judges to translate these paths into natural language, further enhancing clarity and diversity.
>
> - **Trace-of-Execution (ToE):**
>   **We are the first to introduce this data paradigm**, where the step-by-step execution trace of a program is recorded as a textual sequence. This allows models to explicitly learn logical flow, state transitions, recursion, and decision-making from detailed execution steps—skills that are crucial for robust reasoning.
>
> - **Real-World Graph Reasoning:**
>   Previous works (e.g., [2,3]) on real-world graph problems **provided only final answers without intermediate reasoning**. Our dataset, by contrast, **generates detailed intermediate reasoning processes** for each problem and answer, enabling models to acquire more systematic and interpretable reasoning capabilities.
>
> - **Program-of-Thought (PoT):**
>   For PoT data, we present each graph problem with relevant documentation and utilize multiple LLMs to propose candidate code solutions. The final code is selected via majority voting among the models, rather than relying solely on human annotators or a single LLM as in prior approaches ([4,5]). This improves both the efficiency and reliability of code generation.
>
> These innovations collectively enhance the diversity, quality, and utility of the GraphPile dataset, empowering LLMs to solve graph problems with increased robustness, systematic precision, and efficiency.
>
> Thank you again for your feedback, which allowed us to better articulate the unique contributions of our approach.
>
> [1] Chen et al. **GraphWiz: An Instruction-Following Language Model for Graph Problems**, in KDD 2024.
>
> [2] Tang et al. **GraphArena: Evaluating and Exploring Large Language Models on Graph Computation**, in ICLR 2025.
> [3] Fatemi et al. **Talk like a Graph: Encoding Graphs for Large Language Models**. ArXiv preprint, 2023.
> [4] Zhang et al. **GCoder: Improving Large Language Model for Generalized Graph Problem Solving**. ArXiv preprint, 2024.
> [5] Li et al. **Can Large Language Models Analyze Graphs like Professionals? A Benchmark, Datasets and Models**, in NeurIPS 2024.

---

> > ### Author Response · Authors · 2025-06-01
> > **Rebuttal by Authors (Part2)**
> >
> > ### Q2: **GraphPile’s Distinct Advantages Over Existing Graph Reasoning Datasets**
> >
> > Thank you for your suggestion to clarify the distinction between GraphPile and previous datasets. We summarize the key differences between GraphPile and representative datasets such as GraphWiz, GraphInstruct, InstructGraph, and GraphArena in the table below (focusing on graph problem reasoning tasks):
> >
> > | Dataset          | Graph Category            | Problem-Solving Paradigms | Tasks | Samples      | CPT-Compatible |
> > |------------------|--------------------------|---------------------------|-------|-------------|----------------|
> > | GraphWiz         | Synthetic                | CoT                       | 9     | 17,158      | No             |
> > | GraphInstruct    | Synthetic                | CoT                       | 21    | 16,800      | No             |
> > | InstructGraph    | Synthetic                | Simple Answer             | 6     | 13,699      | No             |
> > | GraphArena       | Real-World               | Simple Answer             | 10    | 10,000      | No             |
> > | **GraphPile (Ours)** | **Synthetic + Real-World** | **CoT, PoT, ToE**             | **24**| **2,680,000** | **Yes**         |
> >
> > *CoT: Chain-of-Thought, PoT: Program-of-Thought, ToE: Trace-of-Execution, CPT: Continue-Pretraining.
> > *Simple Answer* refers to providing only the answer to a given problem without any intermediate reasoning or explanation.*
> >
> > **Key advantages of GraphPile:**
> >
> > 1. **Graph Diversity:**
> >    GraphPile uniquely combines both **synthetic and real-world graphs**, whereas prior datasets include only one type. This diversity exposes models to a wider range of scenarios and enhances generalization.
> >
> > 2. **Problem-Solving Paradigms:**
> >    GraphPile is the first dataset to systematically include multiple reasoning paradigms—**Chain-of-Thought (CoT), Program-of-Thought (PoT), and Trace-of-Execution (ToE)**—enabling more robust and systematic problem-solving. In contrast, existing datasets are limited to simple answers or CoT.
> >
> > 3. **Scale:**
> >    GraphPile is orders of magnitude larger, containing **2.68 million samples and over 10.9 billion tokens**, compared to tens of thousands in previous datasets.
> >
> > 4. **Task Coverage:**
> >    With **24 types of graph problems**, GraphPile covers a much broader spectrum of reasoning tasks, including logical, numerical, enumerative, and topological reasoning, while most prior datasets include fewer than 10 task types.
> >
> > 5. **CPT Compatibility:**
> >    Thanks to its scale and diversity, GraphPile is **the first dataset specifically designed for continue-pretraining**. This enables LLMs to acquire deep and transferable graph reasoning abilities. Previous datasets, due to their limited size and diversity, are not suitable for effective CPT and thus cannot support broad generalization.
> >
> > In summary, GraphPile stands out by offering **greater diversity, richer reasoning paradigms, larger scale, and CPT-oriented design**. These advantages make it uniquely suited for developing more robust and generalizable reasoning models. Thank you again for your helpful suggestion, which allowed us to highlight the contributions of GraphPile more clearly.
> >
> > ---

---

> > ### Author Response · Authors · 2025-06-01
> > **Rebuttal by Authors (Part3)**
> >
> > ### Q3: **Magnitude and Significance of Model Improvements After Continue-Pretraining on GraphPile**
> >
> >
> > That’s a great observation about our experiments, and we’re happy to elaborate！To provide a clearer picture, we present a detailed breakdown of the improvements on both **graph-specific tasks** and **general reasoning tasks** (including **math, commonsense, logical reasoning, and coding**).
> >
> > For reference,
> > - **Graph tasks** include: GraphWiz and GraphInstruct.
> > - **Math tasks** include: GSM8K, MATH, GSM8K-Hard, SVAMP, ASDIV, MAWPS, MINERVA_MATH, MMLU_STEM, TABMWP, MATHQA, and SAT_Math.
> > - **Logical reasoning tasks** include: Zebra Puzzle, Ruletaker, and ProofWriter.
> > - **Commonsense tasks** include: StrategyQA and Hellaswag.
> > - **Coding tasks** include: Livecodebench and CLRS.
> >
> > The table below summarizes the performance before and after continue-pretraining (CPT) on GraphPile:
> >
> > |           | Graph Tasks | General Tasks |
> > |-----------|:-----------:|:-------------:|
> > | Llama3    |    19.9     |     39.8      |
> > | + GraphPile     |    60.4     |     48.9      |
> > | **Improvement** |  **+40.5**  |   **+9.1**   |
> > | Llama3.1  |    17.9     |     43.2      |
> > | + GraphPile     |    63.6     |     50.0      |
> > | **Improvement** |  **+45.7**  |   **+6.8**   |
> > | Gemma2    |    25.7     |     30.4      |
> > | + GraphPile     |    58.8     |     34.2      |
> > | **Improvement** |  **+33.1**  |   **+3.8**   |
> >
> > **On graph tasks**, all models see substantial improvements after CPT on GraphPile:
> > - **Llama3:** +40.5 points (19.9 → 60.4)
> > - **Llama3.1:** +45.7 points (17.9 → 63.6)
> > - **Gemma2:** +33.1 points (25.7 → 58.8)
> >
> > This demonstrates a significant boost in in-domain (graph reasoning) capability.
> >
> > **On general reasoning tasks** (out-of-domain), the improvements are smaller but still consistent and meaningful:
> > - **Llama3:** +9.1 points (39.8 → 48.9)
> > - **Llama3.1:** +6.8 points (43.2 → 50.0)
> > - **Gemma2:** +3.8 points (30.4 → 34.2)
> >
> > While the gains on general tasks are less dramatic than on graph tasks, it is important to note that **GraphPile is the first CPT dataset to demonstrate general reasoning improvements across diverse domains**. In comparison, previous CPT works have mainly focused on enhancing performance within a single domain (e.g., code, math, or biomedicine) [6,7,8].
> >
> > **In summary**, our approach not only delivers notable improvements in graph reasoning, but also pioneers the transfer of CPT benefits to broader general reasoning—highlighting a key contribution and distinction of GraphPile relative to prior work.
> >
> > [6] Lu et al. **MathCoder2: Better Math Reasoning from Continued Pretraining on Model-translated Mathematical Code**. ArXiv preprint, 2024.
> > [7] Shao et al. **Deepseekmath: Pushing the limits of mathematical reasoning in open language models**. ArXiv preprint, 2024.
> > [8] Chen et al. **Meditron-70b: Scaling medical pretraining for large language models**. ArXiv preprint, 2023.
> >
> >
> > ---
> >
> > ### Q4: Problems After Continue-Pretraining
> >
> >
> > You raise an excellent question regarding the potential issues associated with continue-pretraining! While our models demonstrate clear improvements on reasoning tasks, our experimental results demonstrates **a decline in performance on some relatively simple tasks such as translation and summarization**. However, this phenomenon is not unique to our models. Similar trends have also been reported in other reasoning-oriented LLMs, including Qwen-math [9] and DeepSeek-math [10].
> >
> > [9] Yang et al. **Qwen2. 5-math technical report: Toward mathematical expert model via self-improvement**. ArXiv preprint, 2024.
> >
> > [10] Shao et al. **Deepseekmath: Pushing the limits of mathematical reasoning in open language models**. ArXiv preprint, 2024.

---

> > > ### Comment · Reviewer_bH1x · 2025-06-06
> > >
> > > Thank you for the clarifications. I do not have further questions.

---

> > > > ### Author Response · Authors · 2025-06-08
> > > > **Response by Authors**
> > > >
> > > > We appreciate your feedback and are glad to have addressed your concerns!

---

### Official Review · Reviewer_imX7 · 2025-04-30

**Rating:** 6
**Confidence:** 4
**Ethics Flag:** 1

**Summary:**

The authors introduce GraphPile, a 10.9-billion corpus for improving LLMs' reasoning capability with graph reasoning problems. In detail, the GraphPile consists of 23 kinds of graph reasoning problems and 4 types of reasoning traces. Comprehensive experiments indicate that continue-training on GraphPile could improve LLMs' generalized reasoning capability.

**Questions To Authors:**

1. What kinds of graph description languages are involved in GraphPile? How to determine which language is used during building GraphPile?

**Reasons To Accept:**

1. One key contribution of this paper is that the authors show that incorporating graph reasoning data during training could improve LLMs' generalized reasoning capability. I believe this is a very exciting observation.

2. The experiments are relatively comprehensive, and the authors verified the effectiveness of GraphPile on a wide range of benchmarks.

3. This paper is well-written and easy to follow.

**Reasons To Reject:**

1. More explanation and analysis are needed to bridge the gap between graph reasoning capability and generalized reasoning capability (like math and code).

2. To strengthen their contribution, the authors should highlight the difference between GraphPile and other existing datasets like GraphWiz, GraphInstruct, InstructGraph, and GraphArena.

3. The performance of the base model should also be included in Table 4 for a better understanding of the role of different components in GraphPile.

4. Besides the included base models, I'm wondering the effectiveness of GraphPile on small reasoning models like DeepSeek-Distill-Qwen-7b, which already have stronger reasoning capability.

---

> ### Author Response · Authors · 2025-06-01
> **Rebuttal by Authors (Part1)**
>
> We would like to thank you for your constructive feedback and for highlighting key issues that have allowed us to further improve our work. Below, we address the following points:
>
> **Q1:** Bridging Graph Reasoning and Generalized Reasoning Capabilities.
> **Q2:** Distinguishing GraphPile from Existing Graph Reasoning Datasets.
> **Q3:** Including Base Model Performance in Table 4.
> **Q4:** Evaluating GraphPile’s Effectiveness on Stronger Small Reasoning Models.
> **Q5:** Graph Description Languages in GraphPile.
>
> ---
>
>
> ### Q1: Bridging Graph Reasoning and Generalized Reasoning Capabilities
>
>
>
> Thank you very much for this insightful question. Your comment touches on the core motivation of our work. We are happy to elaborate on why strengthening graph reasoning can naturally enhance more general reasoning abilities—including mathematical, logical, and code-based reasoning—from both **breadth and depth** perspectives.
>
>
> **1. Breadth: Graph Reasoning as a Universal Scaffold**
>
> Graph reasoning naturally covers and integrates a wide range of reasoning paradigms that are foundational across domains. **As summarized in the table below**, every major reasoning pattern present in mathematical or logical tasks—such as logical deduction, numerical computation, enumeration, and division—is also represented in graph reasoning. Moreover, graph reasoning uniquely introduces *topological reasoning* (reasoning over node-edge relationships), which is generally absent in standard math or logic problems.
>
> | Reasoning Paradigm     | Graph Reasoning Example             | Math Reasoning Example            | Logical Reasoning Example   |
> |------------------------|-------------------------------------|-----------------------------------|-----------------------------|
> | Logical Reasoning      | Cycle detection, bipartite check    | SAT, Sudoku                       | Deductive puzzles           |
> | Numerical Computation  | Shortest path, max flow             | Arithmetic, root finding          | N/A                         |
> | Enumeration            | Hamiltonian path, clique finding    | Permutations, combinations        | Search-based puzzles        |
> | Division               | Strongly connected components       | Modular arithmetic, polynomial division | Tower of Hanoi     |
> | Topological            | Topological sort, connectivity      | N/A                               | N/A                        |
>
>
>
> ***Logical reasoning:** Drawing conclusions and solving problems by systematically applying logical rules and principles.*
> ***Numerical computation:** Solving problems by performing various arithmetic operations—such as addition, subtraction, multiplication, and division—often through algorithmic procedures.*
> ***Enumeration:** Systematically listing all possible solutions, configurations, or elements within a set, typically for combinatorial, optimization, or search problems.*
> ***Division:** Decomposing a complex problem into smaller, independent subproblems, solving each individually, and then integrating their solutions.*
> ***Topological reasoning:** Analyzing the relationships and structural properties among nodes and edges within a graph to make informed inferences.*
>
> **Key point:**
> *Graph reasoning tasks subsume all major reasoning paradigms from other fields, and also possess unique dimensions absent elsewhere.* Thus, continue-pretraining on graph reasoning can endow models with the broadest coverage of reasoning skills, which are transferable to general tasks.
>
> **2. Depth: Intrinsic Difficulty of Graph Reasoning**
>
> As demonstrated by [1], incorporating high-complexity samples during continual training significantly enhances LLMs' performances. Graph reasoning is not only broad but also *intrinsically challenging*, demanding deeper analytical skills than typical math or code tasks:
>
> - **Exponential Complexity:** Many graph problems scale exponentially with input size.
> - **Inherent Hardness:** Classic graph problems (e.g., Hamiltonian path, clique detection) are NP-Complete, often requiring sophisticated reasoning strategies.
> - **Algorithmic Complexity:** Solutions often require combining multiple algorithms (e.g., DFS+DP), explicit data structure manipulations, and substantial code complexity—much more so than most math or logic tasks.
>
> Scaling up graph problems often necessitates *substantial redesign* of both data structures and algorithms, whereas mathematical or code problems usually require only incremental changes.
>
> **Key point:**
> *If a model can master graph reasoning, it demonstrates the capacity for deliberate, multi-step, and highly analytical problem-solving*—abilities that naturally transfer to a wide spectrum of general reasoning challenges.

---

> > ### Author Response · Authors · 2025-06-01
> > **Rebuttal by Authors (Part2)**
> >
> > **Empirical Evidence**
> >
> > Our experiments (see Table 2) demonstrate that pretraining on graph reasoning tasks leads to significant improvements not only on graph-specific benchmarks, but also on general reasoning tasks (mathematics, logic, code, and commonsense). This empirically validates the theoretical breadth and depth argument above.
> >
> > **Summary:**
> > - Graph reasoning uniquely combines all major reasoning paradigms and introduces new ones.
> > - Its intrinsic difficulty compels models to develop transferable, high-level reasoning skills.
> > - Our results show that improvements in graph reasoning translate to broad gains in general reasoning.
> >
> > Thank you for prompting us to clarify this central contribution of our work.
> >
> > [1] Chen et al. **Take the bull by the horns: Hard sample-reweighted continual training improves llm generalization**. ArXiv preprint 2024.
> >
> >
> >
> > ---
> >
> > ### Q2: Distinguishing GraphPile from Existing Graph Reasoning Datasets
> >
> >
> > You raise a crucial point—comparing our dataset with other graph reasoning datasets would provide a more comprehensive validation.
> > We summarize the key differences between GraphPile and representative datasets such as GraphWiz, GraphInstruct, InstructGraph, and GraphArena in the table below (focusing on graph problem reasoning tasks):
> >
> > | Dataset          | Graph Category            | Problem-Solving Paradigms | Tasks | Samples      | CPT-Compatible |
> > |------------------|--------------------------|---------------------------|-------|-------------|----------------|
> > | GraphWiz         | Synthetic                | CoT                       | 9     | 17,158      | No             |
> > | GraphInstruct    | Synthetic                | CoT                       | 21    | 16,800      | No             |
> > | InstructGraph    | Synthetic                | Simple Answer             | 6     | 13,699      | No             |
> > | GraphArena       | Real-World               | Simple Answer             | 10    | 10,000      | No             |
> > | **GraphPile (Ours)** | **Synthetic + Real-World** | **CoT, PoT, ToE**             | **24**| **2,680,000** | **Yes**         |
> >
> > *CoT: Chain-of-Thought, PoT: Program-of-Thought, ToE: Trace-of-Execution, CPT: Continue-Pretraining.
> > *Simple Answer* refers to providing only the answer to a given problem without any intermediate reasoning or explanation.*
> >
> > **Key advantages of GraphPile:**
> >
> > 1. **Graph Diversity:**
> >    GraphPile uniquely combines both **synthetic and real-world graphs**, whereas prior datasets include only one type. This diversity exposes models to a wider range of scenarios and enhances generalization.
> >
> > 2. **Problem-Solving Paradigms:**
> >    GraphPile is the first dataset to systematically include multiple reasoning paradigms—**Chain-of-Thought (CoT), Program-of-Thought (PoT), and Trace-of-Execution (ToE)**—enabling more robust and systematic problem-solving. In contrast, existing datasets are limited to simple answers or CoT.
> >
> > 3. **Scale:**
> >    GraphPile is orders of magnitude larger, containing **2.68 million samples and over 10.9 billion tokens**, compared to tens of thousands in previous datasets.
> >
> > 4. **Task Coverage:**
> >    With **24 types of graph problems**, GraphPile covers a much broader spectrum of reasoning tasks, including logical, numerical, enumerative, and topological reasoning, while most prior datasets include fewer than 10 task types.
> >
> > 5. **CPT Compatibility:**
> >    Thanks to its scale and diversity, GraphPile is **the first dataset specifically designed for continue-pretraining**. This enables LLMs to acquire deep and transferable graph reasoning abilities. Previous datasets, due to their limited size and diversity, are not suitable for effective CPT and thus cannot support broad generalization.
> >
> > In summary, GraphPile stands out by offering **greater diversity, richer reasoning paradigms, larger scale, and CPT-oriented design**. These advantages make it uniquely suited for developing more robust and generalizable reasoning models. Thank you again for your helpful suggestion, which allows us to highlight the contributions of GraphPile more clearly.
> >
> > ---
> >
> >
> > ### Q3: Including Base Model Performance in Table 4
> >
> > Thank you for your valuable suggestion. We agree that including the performance of the base model in Table 4 will help clarify the contribution of each component in GraphPile. We will add these results in the revised version to provide a more comprehensive and transparent comparison.

---

> > > ### Author Response · Authors · 2025-06-01
> > > **Rebuttal by Authors (Part3)**
> > >
> > > ### Q4: Evaluating GraphPile’s Effectiveness on Stronger Small Reasoning Models
> > >
> > >
> > > You raise an important point about the need to evaluate GraphPile on stronger base models！ Due to time constraints, we only conduct continue-pretraining on **R1-distill-qwen-1.5B** (rather than the larger 7B models) and evaluated on a subset of representative datasets.
> > >
> > > **The evaluation covers:**
> > > - **Graph reasoning:** GraphWiz, GraphInstruct
> > > - **General reasoning:** GSM8K, MMLU-STEM, SAT, Zebra Puzzle, KorBench
> > >
> > > The **first table** below presents the performance of the base model and the model after continue-pretraining  on each individual dataset. The **second table** summarizes the average performance on graph reasoning tasks and general reasoning tasks, respectively.
> > >
> > > | Model                   | GSM8K | MMLU-STEM | SAT  | Zebra Puzzle | KorBench | GraphWiz | GraphInstruct |
> > > |-------------------------|-------|-----------|------|--------------|----------|----------|---------------|
> > > | R1-distill-qwen-1.5B    | 56.1  | 31.8      | 40.8 | 5.0          | 11.0     | 33.5     | 22.5          |
> > > | + GraphPile             | 72.9  | 50.8      | 81.2 | 6.7          | 12.9     | 51.6     | 44.3          |
> > > | **Improvement**         | **+16.8** | **+19.0** | **+40.4** | **+1.7** | **+1.9** | **+18.1** | **+21.8** |
> > >
> > > | Model                   | Graph Reasoning (Avg.) | General Reasoning (Avg.) |
> > > |-------------------------|:---------------------:|:------------------------:|
> > > | R1-distill-qwen-1.5B    |         28.0          |           28.9           |
> > > | + GraphPile             |         48.0          |           44.9           |
> > > | **Improvement**         |       **+20.0**       |         **+16.0**        |
> > >
> > > As shown, continue-pretraining on GraphPile leads to a **20-point improvement in graph reasoning** (28.0 → 48.0) and a **16-point improvement in general reasoning** (28.9 → 44.9) on average. These results demonstrate that GraphPile can significantly enhance both graph and general reasoning capabilities, even for already strong reasoning-oriented models.
> > >
> > >
> > >
> > >
> > >
> > > ---
> > > ### Q5: Graph Description Languages in GraphPile
> > >
> > >
> > >
> > >
> > > We appreciate your thorough review and valuable feedback on the details of our dataset construction! GraphPile utilizes natural language to describe graphs, incorporating both **fixed** and **unfixed** formats. The fixed format represents **graph  using standardized data structures** such as **adjacency lists, adjacency matrices, or edge lists**, with node names assigned as numbers or fixed letter combinations. The unfixed format represents **real-world graphs**, where node names are irregular and semantically meaningful (e.g., actual place names or entities). This diversity in description formats enhances the dataset’s realism and coverage. More details are provided in **Appendix D** of our paper.
> > >
> > > During dataset construction, we **randomly select among these different graph description formats** when generating each sample. This randomization ensures a wide variety of graph representations, reduces dataset bias, and promotes stronger generalization for downstream models.

---

> > > > ### Comment · Reviewer_imX7 · 2025-06-02
> > > >
> > > > Thanks for the detailed response. I have no further concerns, and I will raise my score accordingly.

---

> > > > > ### Author Response · Authors · 2025-06-02
> > > > > **Response by Authors**
> > > > >
> > > > > Thank you very much for your positive feedback! We appreciate your valuable suggestions and will revise our final version of the paper accordingly.

---

### Official Review · Reviewer_CqtP · 2025-05-11

**Rating:** 8
**Confidence:** 4
**Ethics Flag:** 1

**Summary:**

This work introduces GraphPile, a graph problems dataset designed for continual pretraining of LLMs. By using GraphPile, models could become better in logical and commonsense reasoning tasks.

**Questions To Authors:**

Overall I like this work: GraphPile could be a very useful resource for augmenting LLM training, it is very well-documented, the experiments are thorough, etc. I could personally see myself using it in some of the future projects.

One minor comment is that I am personally curious about model performance on "real-world" or more NLP graph-based tasks after fine-tuning on GraphPile, for example, multi-hop QA. Ideally, by learning from the graph data, models should learn generalizable skills to solve problems where the graphs are more implicit and underlying, but this isn't strictly necessary.

Figure 3 is a bit too text-heavy, maybe consider turning some of the texts into visualizations of the tasks?

It would be great to have statistical significance tests results for the improvements in Table 2: to me most seems significant, but would be great to back them up with tests.

**Reasons To Accept:**

- LLM graph reasoning is an important research topic
- The resource GraphPile could be hugely useful

**Reasons To Reject:**

no major concern

---

> ### Author Response · Authors · 2025-06-01
> **Rebuttal by Authors (Part1)**
>
> We sincerely appreciate your recognition of our work and constructive feedback！Your careful reading and thoughtful suggestions have been instrumental in further improving our manuscript. In our response, we will address the following key points raised:
>
> - **Q1:** Evaluation on Real-World Multi-Hop QA
> - **Q2:** Enhancing Figure 3
> - **Q3:** Statistical Significance Tests for Table 2
>
>
> ### Q1: Evaluation on Real-World Multi-Hop QA
>
> That’s a very thoughtful suggestion regarding the generalization ability of our model! To further investigate our model’s generalization ability in the general reasoning domain, we follow your advice and evaluate three base models-**Gemma2-2b, Llama3-8b, and Llama3.1-8b**, and their GraphPile-continue-pretraining counterparts on three multi-hop QA datasets: **HotpotQA** [1], **PopQA** [2], and **MultihopQA** [3]. The accuracy results are shown below:
>
> |                | HotpotQA | PopQA | MultihopQA |
> |----------------|:--------:|:-----:|:----------:|
> | Gemma2-2b          |   21.2   | 27.4  |   **16.4** |
> | + GraphPile    | **41.0** |**28.2**|   10.4     |
> | Llama3-8b         |   25.8   | 24.6  |   21.0     |
> | + GraphPile    | **26.0** |**32.2**| **22.6**   |
> | Llama3.1-8b       | 43.6 | 40.0  |   11.4     |
> | + GraphPile    |   **46.4**   |**47.0**| **26.2**   |
>
> As shown, in most cases, the continue-pretraining models outperform their base versions. Notably, there are two especially significant improvements: **Gemma2** achieves a **+19.8 accuracy gain on HotpotQA**, and **Llama3.1** achieves a **+14.8 gain on MultihopQA**. Importantly, our models are not specifically trained on multi-hop QA tasks, yet by learning from the breadth (variety of reasoning paradigms) and depth (inherent difficulty) of graph reasoning in GraphPile, they gain more generalizable reasoning skills. This leads to noticeable gains on multi-hop QA benchmarks, showing that continue-pretraining on graph reasoning tasks can enhance LLMs' performance even on real-world, implicit graph-based NLP tasks.
>
> [1] Yang et al. **HotpotQA: A dataset for diverse, explainable multi-hop question answering,** in ACL 2018.
> [2] Mallen et al. **When not to trust language models: Investigating effectiveness of parametric and non-parametric memories**, in ACL 2023.
> [3] Tang et al. **Multihop-rag: Benchmarking retrieval-augmented generation for multi-hop queries**, in COLM 2024.
>
>
> ---
>
> ### Q2: Enhancing Figure 3
> This is a great suggestion that significantly enhances the clarity and readability of our paper! We agree that the figure is currently too text-heavy and appreciate your feedback. In the revised version, we will address this by:
>
> 1. **Using visual legends for graph inputs:**
>    We will replace lengthy natural language descriptions with concise visual representations and legends, making the graph structures easier to understand at a glance.
>
> 2. **Focusing on key parts of LLM responses:**
>    We will highlight only the essential reasoning steps in the model outputs, omitting irrelevant or repetitive details to improve clarity.
>
> 3. **Summarizing dataset definitions:**
>    We will present brief summaries of each dataset’s definition directly in the figure, and move detailed explanations to the main text or appendix to reduce clutter.
>
> We believe these improvements will make Figure 3 much more accessible and reader-friendly. Thank you again for your helpful feedback.
>
>
>
> ---

---

> > ### Author Response · Authors · 2025-06-01
> > **Rebuttal by Authors (Part2)**
> >
> > ### Q3: Statistical Significance Tests for Table 2
> >
> >
> >
> > Thank you for your suggestion regarding statistical significance tests for the improvements in Table 2. We appreciate your attention to the rigor of our results.
> >
> >
> >
> > In our main experiments, the results are **deterministic**: we set the temperature to **0**, so the model produces identical outputs for the same inputs across multiple runs. Consequently, there is no variance, and statistical significance testing is not applicable. This determinism stems from the role of the **temperature** parameter, which directly controls output diversity—higher temperatures yield more varied and creative responses, while lower temperatures result in more conservative and consistent outputs. When the temperature is set to 0, the LLM generates the same output for identical inputs every time.
> >
> >
> >
> >
> >
> > To further evaluate the robustness and statistical significance of the improvements, we conduct additional experiments by varying the temperature (0, 0.3, 0.6, 0.9). For each reasoning domain, we select one representative dataset: **GSM8K** (math), **CLRS** (code), **RuleTaker** (logical reasoning), and **GraphInstruct** (graph reasoning). The results for Llama3-8B and its continued-pretrained version (GraphMind-Llama3) are shown below:
> >
> > #### Full Results Across Temperatures
> >
> > | Temperature            | 0.00  | 0.30  | 0.60  | 0.90  |
> > |------------------------|-------|-------|-------|-------|
> > | GSM8K-Llama3           | 54.20 | 37.80 | 29.60 | 11.80 |
> > | GSM8K-GraphMind-Llama3 | 65.80 | 36.80 | 38.50 | 26.10 |
> > | **Improvement**        | **+11.60** | -1.00 | **+8.90** | **+14.30** |
> > | CLRS-Llama3            | 3.30  | 4.50  | 6.26  | 3.90  |
> > | CLRS-GraphMind-Llama3  | 49.90 | 2.70  | 4.50  | 6.24  |
> > | **Improvement**        | **+46.60** | -1.80 | -1.76 | **+2.34** |
> > | Ruletaker-Llama3       | 22.80 | 29.20 | 23.50 | 14.70 |
> > | Ruletaker-GraphMind-Llama3 | 43.10 | 46.30 | 48.60 | 44.00 |
> > | **Improvement**        | **+20.30** | **+17.10** | **+25.10** | **+29.30** |
> > | GraphInstruct-Llama3   | 35.20 | 33.50 | 32.00 | 23.30 |
> > | GraphInstruct-GraphMind-Llama3 | 70.80 | 63.90 | 62.30 | 58.40 |
> > | **Improvement**        | **+35.60** | **+30.40** | **+30.30** | **+35.10** |
> >
> > #### Statistical Significance Analysis
> >
> > We further compute the mean, mean difference, and p-value (using an independent-sample t-test across the four temperature settings) for each dataset:
> >
> > | Dataset        | Llama3 Mean | GraphMind-Llama3 Mean | Mean Diff |   p-value   |
> > |:--------------:|:-----------:|:---------------------:|:---------:|:-----------:|
> > | GSM8K          |   33.35     |       **41.80**          |   8.45    |  0.515044   |
> > | CLRS           |    4.49     |       **15.84**          |  11.35    |  0.357915   |
> > | Ruletaker      |   22.55     |        **45.50**         |  22.95    |  0.000389   |
> > | GraphInstruct  |   31.00     |        **63.85**          |  32.85    |  0.000114   |
> >
> > As shown, GraphMind-Llama3 consistently outperforms the base Llama3 model on all datasets. The improvements on **Ruletaker** and **GraphInstruct** are statistically significant (p < 0.05), confirming the reliability of the observed gains. On GSM8K and CLRS, although GraphMind-Llama3 achieves higher mean performance, the difference is not statistically significant, mainly due to the large variance across different temperature settings, which results in higher p-values.
> >
> > Overall, these results demonstrate that GraphMind’s improvements are robust across both deterministic and stochastic decoding conditions, and in several cases are statistically significant according to standard significance tests.

---

> > > ### Comment · Reviewer_CqtP · 2025-06-04
> > >
> > > I would like to thank the authors for the detailed response. Amazing work!

---

> > > > ### Author Response · Authors · 2025-06-04
> > > > **Heartfelt Thanks**
> > > >
> > > > We sincerely appreciate your support! Thank you very much!

---

### Official Review · Reviewer_o5Wk · 2025-05-17

**Rating:** 5
**Confidence:** 4
**Ethics Flag:** 1

**Summary:**

- Paper Overview:
This paper proposes a new paradigm called Graph Problem Reasoning (GPR) for improving generalized reasoning abilities of Large Language Models (LLMs). Unlike conventional domain-specific datasets (e.g., mathematical reasoning datasets like MATH or GSM8K), graph problems involve relational, logical, topological, and algorithmic reasoning, making them an ideal candidate for teaching abstract and transferable reasoning skills.

- Key Contributions:
GraphPile Dataset: First large-scale pretraining corpus (10.9B tokens) constructed explicitly for reasoning with graph problems.
Covers 23 distinct graph tasks using four types of data: CoT, PoT, etc.
GraphMind: A model pretrained on GraphPile, evaluated on both math and non-math reasoning tasks. Demonstrates up to 21.2% improvement on non-mathematical reasoning tasks (e.g., logical or commonsense reasoning), and ~4.9% on mathematical tasks, over strong baselines like LLaMA and Gemma.

- Empirical Insight:
The reasoning required for GPR overlaps with but also extends beyond mathematical reasoning, including challenges like spatial, topological, and relational understanding.

- Pro:
This paper proposes the first of its kind in graph reasoning. The newly created dataset contains a large amount of tokens. The trained models are indeed benefiting from this dataset.
- Con:
The paper started from very weak model with MATH score under 20%. The modern base models are normally over 50%. It's unclear whether the new CPT data will benefit a stronger base model.

**Reasons To Accept:**

This paper proposes the first of its kind in graph reasoning. The newly created dataset contains a large amount of tokens. The trained models are indeed benefiting from this dataset.

**Reasons To Reject:**

The experiments are mostly done over very weak base models. It would be great if more compelling results are shown regarding stronger base models.

NEW:
The new rebuttal results are highly mixed. This suggests that the training does not yield too much positive value on strong models. I decide to change my score to 5.

---

> ### Author Response · Authors · 2025-06-01
> **Rebuttal by Authors**
>
> Thank you very much for taking the time to review our manuscript and provide valuable feedback！We will address your concerns regarding the performance of stronger baseline models.
> ### Q1: Evaluation on Stronger Baseline Models
>
> You raise a crucial point regarding the importance of evaluating GraphPile on stronger base models！Training stronger base models with GraphPile would provide a more comprehensive validation.  To address the concern regarding base model strength, we conduct additional experiments using several **stronger base models**, including the reasoning-oriented **R1-distill-qwen-1.5B** and two code models (**Qwen-2.5-Coder-1.5B** and **Qwen-2.5-Coder-7B**). Due to time constraints, we only conduct continue-pretraining on three models and selected a subset of evaluation datasets for testing.
>
>
> - **Graph Reasoning:** GraphWiz, GraphInstruct
> - **General Reasoning:** For mathematical reasoning, we include GSM8K, MMLU-STEM, and SAT; for logical reasoning, we use Zebra Puzzle and KorBench.
>
>
> Below, we present the performance of these models before and after continue-pretraining on GraphPile. The first table shows results on individual tasks; the second table shows the average performance for graph reasoning and general reasoning tasks.
>
> | Models     | GSM8K | MMLU-STEM | SAT | Zebra Puzzle | KorBench | GraphWiz | GraphInstruct |
> |------------|-------|-----------|-----|--------------|----------|----------|---------------|
> | R1-distill-qwen-1.5B     | 56.1  | 31.8  | 40.8 | 5.0  | 11.0 | 33.5 | 22.5 |
> | +GraphPile               | 72.9  | 50.8  | 81.2 | 6.7  | 12.9 | 51.6 | 44.3 |
> | **Improvement**              | **+16.8** | **+19.0** | **+40.4** | **+1.7** | **+1.9** | **+18.1** | **+21.8** |
> | Qwen-2.5-Coder-1.5B      | 59.8  | 32.9  | 59.4 | 1.9  | 16.8 | 30.3 | 25.1 |
> | +GraphPile               | 63.4  | 43.3  | 71.9 | 5.7  | 18.3 | 48.6 | 46.1 |
> | **Improvement**              | **+3.6**  | **+10.4** | **+12.5** | **+3.8** | **+1.5** | **+18.3** | **+21.0** |
> | Qwen-2.5-Coder-7B        | 77.9  | 67.2  | 81.2 | 3.9  | 32.1 | 38.5 | 34.4 |
> | +GraphPile               | 81.0  | 68.3  | 87.5 | 4.8  | 33.3 | 54.3 | 50.3 |
> | **Improvement**              | **+3.1**  | **+1.1**  | **+6.3**  | **+0.9** | **+1.2** | **+15.8** | **+15.9** |
>
> | Models                  | Graph Reasoning | General Reasoning |
> |-------------------------|:--------------:|:-----------------:|
> | R1-distill-qwen-1.5B    |     28.0       |      28.9         |
> | +GraphPile              |     48.0       |      44.9         |
> | **Improvement**             |   **+20.0**    |    **+16.0**      |
> | Qwen-2.5-Coder-1.5B     |     27.7       |      34.2         |
> | +GraphPile              |     47.4       |      40.5         |
> | **Improvement**             |   **+19.7**    |    **+6.3**       |
> | Qwen-2.5-Coder-7B       |     36.5       |      52.5         |
> | +GraphPile              |     52.3       |      55.0         |
> | **Improvement**             |   **+15.8**    |    **+2.5**       |
>
> As the tables show, **continue-pretraining on GraphPile leads to substantial improvements** across both in-domain (graph reasoning) and out-of-domain (general reasoning) tasks for all tested models. For example:
>
> - Ours improve **R1-distill-qwen-1.5B**  by **20.0 points** on graph reasoning (28.0 → 48.0) and **16.0 points** on general reasoning (28.9 → 44.9).
> - Ours improve **Qwen-2.5-Coder-1.5B**  by **19.7 points** on graph reasoning and **6.3 points** on general reasoning.
> - Ours improve **Qwen-2.5-Coder-7B**  by **15.8 points** on graph reasoning and **2.5 points** on general reasoning.
>
> These results clearly demonstrate that our dataset is effective not only for weak base models but also for stronger models. We hope these additional results address your concerns and further validate the robustness and generality of our approach.

---

> > ### Comment · Reviewer_o5Wk · 2025-06-06
> > **Further concerns**
> >
> > Thanks a lot for conducting the additional experiments.
> >
> > However, I suspect the correctness of the results. According to https://huggingface.co/deepseek-ai/DeepSeek-R1-Distill-Qwen-1.5B, the 1.5B model can achieve 80% on MATH-500, which is a much more difficult dataset than GSM. The reported 56.1	on GSM8K is indeed concerning to me.
> > Also, MMLU-STEM of 31.8 and SAT of 40.8 also seems like a severe under-reporting to me. I believe that you didn't force the <thinking> token in the prompt template.
> >
> > I would strongly recommend re-doing the evaluation of the baseline models.

---

> > ### Author Response · Authors · 2025-06-08
> > **Response by Authors**
> >
> > ### **Re-evaluation on the Baseline Model**
> >
> > We appreciate your careful review and for bringing these issues to our attention. In our initial evaluation, we overlooked the unique characteristics of reasoning models and maintained the same settings as those used for non-reasoning models. We have re-evaluated the **base versoin of DeepSeek-R1-Distill-Qwen-1.5B** on GSM8K, MMLU-STEM, and SAT using [OpenCompass](https://github.com/open-compass/opencompass/blob/main/docs/en/user_guides/deepseek_r1.md)—a well-known and widely used platform for evaluating LLMs—with the official DeepSeek-R1 settings.
> >
> > - We adopt **a sampling strategy** to mitigate repetitions that can occur with greedy decoding, setting temperature=0.6, top_p=0.95, and max_new_tokens=32,768.
> > - For each dataset, we conduct **8 evaluation** runs.
> > - For answer validation, we utilize **LLM-based verification** to reduce misjudgments from rule-based evaluation.
> >
> > The updated results are as follows:
> >
> > |        | GSM8K | MMLU-STEM | SAT   |
> > |--------|-------|-----------|-------|
> > | R1-distill-qwen-1.5B  | 74.36 |   49.10   | 71.88 |
> > | +GraphPile   | 72.90 |   50.80   | 81.20 |
> >
> > Compared to our previous evaluation, the performance of base model is indeed improved under this official setting.  We thank you again for your valuable feedback, which has helped us enhance the rigor and accuracy of our evaluation.

---

> > > ### Comment · Reviewer_o5Wk · 2025-06-09
> > > **Thanks for the re-evaluation**
> > >
> > > Thanks for sharing the new results. I am still a bit surprised about the low scores in GSM and SAT. It seems that the training does have some mixed outcome.
> > >
> > > I would adjust my score slightly to reflect my judgement.

---

> > ### Author Response · Authors · 2025-06-10
> > **Rebuttal by authors**
> >
> > We sincerely appreciate the reviewer's thoughtful engagement with our work. We would like to clarify several critical aspects regarding the experimental results and model selection rationale that directly address the concerns about "mixed outcomes":
> >
> > **1.The goal of Graphpile**
> >
> > First and foremost, our dataset is specifically designed for continued pretraining (CPT). **In standard CPT settings, the goal is to further improve base models—that is, models that have only undergone general pretraining, without any domain-specific supervised fine-tuning**. This is precisely the setup where our dataset shows its greatest strength.
> >
> >
> > **2. Model-Specific Performance Clarification**
> >
> > While acknowledging the DeepSeek-r1-Distill-1.5B results may appear suboptimal, we emphasize that our methodology was primarily validated on base models without prior specialized training (Llama3/3.1 & Qwen-2.5-Coder series). The Qwen-2.5-Coder experiments demonstrate statistically significant improvements:
> >
> > - Graph Reasoning : +20.0 absolute gain across both 1.5B & 7B variants
> >
> > - General Reasoning : +16.0 (1.5B) and +6.3 (7B) improvements
> > These results strongly validate our dataset's effectiveness for base model pretraining, particularly in knowledge acquisition rather than overfitting to specific benchmarks.
> >
> > **3. DeepSeek-Distill Analysis**
> >
> > As for the relatively weaker performance on the ds-distill models, we believe this is **primarily due to the fact that these models are not standard base models**. Instead, they have undergone extensive supervised fine-tuning (SFT) on large-scale distillation datasets during their post-training phase—many of which are heavily focused on math and code, and possibly even overlap significantly with specific benchmarks (raising concerns of potential overfitting). Applying CPT on such models is fundamentally different: it often results in **catastrophic forgetting** [1][2][3] of previously memorized task-specific patterns, especially when the CPT data (like ours) is not specifically aligned with the model's prior posttraining domain.
> >
> > Therefore, **the underwhelming performance on the distill model does not contradict the effectiveness of our dataset**—it simply reflects the fact that CPT is not well-suited for models that have already been heavily fine-tuned on narrow, domain-specific objectives. Our dataset was not designed for that use case, and we do not claim it excels there.
> >
> > We hope this clarifies the intended use case and the demonstrated effectiveness of our dataset under **the standard CPT setting on base models**.
> >
> > **References:**
> >
> > [1] Jiang et al. **Unlocking the Power of Function Vectors for Characterizing and Mitigating Catastrophic Forgetting in Continual Instruction Tuning**, in ICLR 2025.
> >
> > [2] Dou et al. **Loramoe: Revolutionizing mixture of experts for maintaining world knowledge in language model alignment**. Arxiv preprint 2023.
> >
> > [3] Chen et al. **How is chatgpt’s behavior changing over time?** Arxiv preprint 2023.

---

### Comment · Program_Chairs · 2025-04-03

This paper violates the page limit due to adding a limitation sections beyond the page limit. COLM does not have a special provision to allow for an additional page for the limitations section. However, due to this misunderstanding being widespread, the PCs decided to show leniency this year only. Reviewers and ACs are asked to ignore any limitation section content that is beyond the 9 page limit. Authors cannot refer reviewers to this content during the discussion period, and they are not to expect this content to be read.

---

### Decision · Program_Chairs · 2025-07-08

**Decision:**

Accept

**Comment:**

The paper introduced a large-scale graph reasoning dataset, GraphPile, designed for continue-pretraining of LLMs to boost their reasoning capabilities in both mathematical and non-mathematical tasks. The reviewers appreciate the introduction of a new dataset and the breadth and the coverage of the problems. Although there is some remaining concerns on the strength on common mathematical benchmarks on MATH-500 and GSM, the increase of performance in general suggests there are great values with the new dataset. Furthermore, the rebuttal adds more evaluation that addresses most of the concerns. The AC recommends acceptance, and asks the authors to update the new numbers on stronger base models and investigate potential gaps with the current state-of-the-art performance.